# Cannabinoids in Medicine: A Multifaceted Exploration of Types, Therapeutic Applications, and Emerging Opportunities in Neurodegenerative Diseases and Cancer Therapy

**DOI:** 10.3390/biom13091388

**Published:** 2023-09-14

**Authors:** Victor Voicu, Felix-Mircea Brehar, Corneliu Toader, Razvan-Adrian Covache-Busuioc, Antonio Daniel Corlatescu, Andrei Bordeianu, Horia Petre Costin, Bogdan-Gabriel Bratu, Luca-Andrei Glavan, Alexandru Vlad Ciurea

**Affiliations:** 1Pharmacology, Toxicology and Clinical Psychopharmacology, “Carol Davila” University of Medicine and Pharmacy in Bucharest, 020021 Bucharest, Romania; victor.voicu@yahoo.com; 2Medical Section within the Romanian Academy, 010071 Bucharest, Romania; 3Neurosurgery Department, Emergency Clinical Hospital Bagdasar-Arseni, 041915 Bucharest, Romania; 4Department of Neurosurgery, “Carol Davila” University of Medicine and Pharmacy, 020021 Bucharest, Romania; razvan-adrian.covache-busuioc0720@stud.umfcd.ro (R.-A.C.-B.); antonio.corlatescu0920@stud.umfcd.ro (A.D.C.); andrei.bordeianu@stud.umfcd.ro (A.B.); horia-petre.costin0720@stud.umfcd.ro (H.P.C.); bogdan.bratu@stud.umfcd.ro (B.-G.B.); luca-andrei.glavan0720@stud.umfcd.ro (L.-A.G.); prof.avciurea@gmail.com (A.V.C.); 5Department of Vascular Neurosurgery, National Institute of Neurology and Neurovascular Diseases, 077160 Bucharest, Romania; 6Neurosurgery Department, Sanador Clinical Hospital, 010991 Bucharest, Romania

**Keywords:** cannabinoids, neurodegenerative diseases, phytocannabinoids, antitumor properties, endocannabinoids, cannabinoid receptors

## Abstract

In this review article, we embark on a thorough exploration of cannabinoids, compounds that have garnered considerable attention for their potential therapeutic applications. Initially, this article delves into the fundamental background of cannabinoids, emphasizing the role of endogenous cannabinoids in the human body and outlining their significance in studying neurodegenerative diseases and cancer. Building on this foundation, this article categorizes cannabinoids into three main types: phytocannabinoids (plant-derived cannabinoids), endocannabinoids (naturally occurring in the body), and synthetic cannabinoids (laboratory-produced cannabinoids). The intricate mechanisms through which these compounds interact with cannabinoid receptors and signaling pathways are elucidated. A comprehensive overview of cannabinoid pharmacology follows, highlighting their absorption, distribution, metabolism, and excretion, as well as their pharmacokinetic and pharmacodynamic properties. Special emphasis is placed on the role of cannabinoids in neurodegenerative diseases, showcasing their potential benefits in conditions such as Alzheimer’s disease, Parkinson’s disease, Huntington’s disease, and multiple sclerosis. The potential antitumor properties of cannabinoids are also investigated, exploring their potential therapeutic applications in cancer treatment and the mechanisms underlying their anticancer effects. Clinical aspects are thoroughly discussed, from the viability of cannabinoids as therapeutic agents to current clinical trials, safety considerations, and the adverse effects observed. This review culminates in a discussion of promising future research avenues and the broader implications for cannabinoid-based therapies, concluding with a reflection on the immense potential of cannabinoids in modern medicine.

## 1. Introduction

### 1.1. Background on Cannabinoids

The burgeoning field of research surrounding Cannabis sativa has identified Δ9-tetrahydrocannabinol (THC) as one of its key active compounds. This compound, along with a plethora of related molecules, both phytochemical and synthetic, exhibits an affinity for specific neuronal binding sites, contributing to an altered mood and perceptions. These binding locations are particularly abundant in neural structures such as the substantia nigra, hippocampus, and cerebellum. One of the clinical manifestations associated with excessive cannabis consumption is cannabinoid hyperemesis syndrome (CHS), a condition characterized by recurrent episodes of severe vomiting. This syndrome is exacerbated by the consumption of high doses of cannabinoids and has imposed a growing burden on healthcare systems, particularly in the United States. Despite the increasing prevalence of CHS, the medical community has yet to reach a consensus on optimal treatment strategies [1,2].

Beyond their well-documented medicinal applications, cannabinoids have been the subject of intensive investigations aimed at exploring their therapeutic potential for a variety of medical conditions, including pain, addiction, obesity, and inflammation, among others. Recent discoveries have expanded our understanding of the pharmacology of cannabinoids by revealing the existence of non-CB1 and non-CB2 orphan G-protein-coupled receptors such as GPR18, GPR55, and GPR119. These receptors operate in conjunction with the established CB1 and CB2 receptors but have unique characteristics, including allosteric binding and biased signaling, which could lead to distinct functional outcomes. A particularly intriguing line of inquiry has revealed the presence of CB1 receptors within the mitochondria of striated and cardiac muscles, implicating them in the modulation of intramitochondrial signaling and respiratory processes [3,4,5].

### 1.2. Importance of Endogenous Cannabinoids

The endocannabinoid system (ECS) represents a complex neuromodulatory network that is of paramount significance to the central nervous system (CNS), synaptic plasticity, and adaptive responses to both endogenous and environmental stimuli. Comprising cannabinoid receptors (CBRs), endogenously synthesized cannabinoids (endocannabinoids), and enzymatic pathways for their synthesis and degradation, the ECS serves as a critical mechanism for neuromodulation. While CB1 cannabinoid receptors are the most commonly implicated receptors in these interactions, other receptors such as CB2, transient receptor potential channels, and peroxisome proliferator-activated receptors also participate [5,6].

Two endogenous cannabinoids that have attracted significant academic attention are 2-arachidonoyl glycerol (2-AG) and arachidonoyl ethanolamide (anandamide). Despite sharing molecular similarities, these cannabinoids diverge in their synthetic and degradative enzymatic pathways, leading to distinct physiological and pathophysiological roles [7,8].

The societal ubiquity of cannabis consumption has fueled a wealth of research into the physiological and pathophysiological functions of endocannabinoids. Marijuana’s prevalence as a widely consumed substance in Western societies contributed to the discovery of the ECS and elucidated its involvement in a plethora of physiological processes. This intricate system comprises G-protein-coupled CBRs that are activated by lipid mediators, commonly referred to as endocannabinoids (eCBs). These eCBs are not only synthesized from cannabis but also encompass a variety of biochemical constituents including precursors, enzymes, and transporters. Research has revealed an extensive distribution of components of the ECS throughout various bodily regions and organs, underscoring its fundamental role in physiology and the potential for targeted interventions for a range of human ailments [9,10,11,12].

Historically, Cannabis sativa (marijuana) has been employed to stimulate appetite, but rigorous scientific scrutiny of its molecular mechanisms gained momentum following the identification of THC in the late 1960s. Although marred by societal disapproval due to misuse, empirical evidence has highlighted the therapeutic potential of marijuana and its derivatives. Specifically, they have been found to enhance appetite for sweet foods. The elucidation of distinct CBRs and their endogenous ligands has provided a robust physiological framework for understanding the myriad biological effects mediated by marijuana and other cannabinoids [13,14].

Recent advancements in the field have illuminated the existence of a plethora of naturally occurring compounds that serve as binding partners to CBRs. eCBs bear a functional resemblance to endorphins and have been detected in a range of mammalian species, including humans. Notably, eCBs have been identified in a diverse array of tissues such as the CNS, peripheral nerves, and reproductive and immune organs like the uterus, leukocytes, spleen, and testicles. Anandamide, one of the earliest-discovered eCBs, is present in notably high concentrations in uterine tissue [15]. This suggests a pivotal role in reproductive processes, a notion corroborated via extensive investigations. Empirical studies have revealed anandamide’s crucial involvement in orchestrating implantation processes. A diminished enzymatic activity responsible for the breakdown of anandamide has been frequently associated with early pregnancy loss [16,17]. The growing body of scientific literature on eCBs has notably concentrated on the study of anandamide, reaffirming its significance in both physiological and pathophysiological contexts.

### 1.3. Overview of Neurodegenerative Diseases and Cancer

THC demonstrates interactions with CBRs, chiefly CB1 and CB2, which are naturally activated by eCBs. The compound plays a multifaceted role in various physiological and pathological domains, including the modulation of the release of neurotransmitters, the regulation of pain perception, and the functioning of the cardiovascular, digestive, and hepatic systems. Nonetheless, THC’s psychotropic effects, which are mainly mediated via the activation of CB1 receptors in the brain, have considerably restricted its clinical applicability. Contrastingly, the cannabis plant is replete with cannabinoids that exhibit minimal to no psychotropic activity, many of which have demonstrated therapeutic potential surpassing that of THC. Among these, cannabidiol (CBD) has gained prominence for its prospective utility in treating conditions such as inflammation, diabetes, cancer, affective disorders, and neurodegenerative diseases. Another cannabinoid, D9-tetrahydrocannabivarin (THCV), shows promise in addressing issues like epilepsy and obesity [18,19].

Neurological disorders, inclusive of neurodegenerative diseases and traumatic brain injuries, pose considerable challenges to healthcare due to their impact on cognitive, motor, and behavioral functions. While diverse therapeutic strategies have been explored, none have provided definitive results. However, accumulated evidence indicates that cannabinoids may offer a novel pathway for treatment. Research elucidates the in vivo potential of both natural and synthetic cannabinoids in ameliorating cognitive decline and motor impairments. Animal models have demonstrated the efficacy of cannabinoids in enhancing neurobehavioral function, improving working memory, and reducing neurological deficits through mechanisms such as modulating inflammation, mitigating edema, and preserving the neuronal structure [20,21].

CBD’s antioxidative properties have revealed its potential in combating neurodegenerative and cardiovascular disorders. Moreover, animal studies have showcased CBD’s anticancer properties. The co-administration of THC with radiation therapy has also been observed to induce higher rates of autophagy and apoptosis in cancer cells. The National Cancer Institute acknowledges the therapeutic potential of Cannabis sativa, particularly THC and CBD, in alleviating various symptoms associated with cancer, including pain, appetite loss, nausea, and anxiety. CBD’s complex pharmacological profile allows it to act as an adaptogen and modulator, interacting intricately with the receptor proteins CB1 and CB2, among other sites [22].

CBD is increasingly being recognized for its potential as an immunomodulatory entity. Empirical studies substantiate its efficacy in engendering immunosuppression against non-infectious inflammatory conditions, such as inflammatory bowel disease, rheumatoid arthritis, and neurodegenerative disorders. Moreover, CBD has exhibited immunoprotective qualities against viral infections, including COVID-19. Its interactions with an array of cellular targets and signaling pathways have been found to induce specific anti-cancer responses, which is in alignment with its principal role in ECS-mediated homeostasis [23].

Case reports in the medical literature affirm the therapeutic viability of cannabinoids derived from Cannabis sativa. However, the clinical adoption of these compounds is often hindered by the psychotropic side effects that are predominantly attributable to THC. Advancements in the understanding of the ECS, including the discovery of new receptors, ligands, and mediators, have facilitated the exploration of novel therapeutic avenues that could mitigate the adverse psychotropic effects associated with certain plant constituents. Such scientific innovations have catalyzed the development of FDA-approved medications that are revolutionizing contemporary medical treatment modalities. For instance, Nabiximols, an FDA-sanctioned amalgam of THC and non-psychoactive CBD, has demonstrated utility in alleviating the pain and spasticity related to multiple sclerosis [24,25]. Additionally, DRO and Nabilone have gained FDA approval for their effectiveness in countering chemotherapy-induced nausea and vomiting in cancer patients. Notably, DRO has also secured regulatory endorsement for its role in managing anorexia among AIDS patients [26,27,28,29].

### 1.4. Significance of Studying Cannabinoid Effects

In a comprehensive meta-analysis incorporating 211 studies, the binding affinities of cannabinoid receptor ligands at human (Hs) and rat (Rn) CB1 and CB2 receptors were examined. Methodologies in line with the Cochrane procedures guided this nonclinical investigation. Meta-regression techniques were utilized to identify data variances due to methodological factors. The Ki values for THC exhibited discernible differences between HsCB1 and RnCB1. The Kd values for CP55,940 and WIN55,212-2 also displayed significant discrepancies between HsCB1 and RnCB1, as well as between HsCB1 and HsCB2. Moreover, SR141716A exhibited affinity to both sets of CBRs [30].

Another exhaustive analysis considered 91 publications consisting of 104 individual studies with 9958 participants. These studies ranged from randomized controlled trials (RCTs) to observational research and covered a variety of pain-related conditions including neuropathic pain, fibromyalgia, and chronic non-cancer pain (CNCP). Pooled event rates (PERs) revealed that cannabinoids were statistically more effective than a placebo for achieving a 30% reduction in pain, but no significant difference was found for a 50% reduction in pain [31]. 

A discernible uptick has been observed in the usage of synthetic cannabinoid products among adolescents. A study concentrated on the self-reported psychoactive and somatic ramifications of synthetic cannabinoid use among adolescents. Notably, all participants indicated experiencing euphoria and memory alterations. A significant majority, 82%, also reported negative mood shifts. The concurrent use of marijuana and alcohol was noted by 91% of the subjects. Intriguingly, a robust correlation was observed between the frequency of synthetic cannabinoid use and the number of other drugs consumed (r = 0.896, *p* < 0.05). Consequently, the study concluded that adolescent users of synthetic cannabinoids report substantial psychoactive effects [32]. 

Previous research has shown that the stimulation of CB1 receptors affects both motility and food intake in rodent models and also has implications for human gastrointestinal (GI) function; however, specific effects on human GI transit times and sensations of fullness remain undetermined. To shed light on this, a double-blind, randomized study involving 30 healthy volunteers assessed the effects of DRO versus a placebo through a series of diagnostic tests, including the Ensure^®^ Satiation test and scintigraphic transit testing [33]. In summary, the study posits that the ECS within the human gastrointestinal tract can be modulated by the non-selective cannabinoid receptor (CBR) agonist DRO to decelerate gastric emptying. The study further advocates for subsequent investigations involving both selective and non-selective cannabinoid antagonists to substantiate these initial findings. Owing to preliminary evidence indicating gender-specific variations in gastric emptying and fasting gastric volume in response to the acute administration of DRO, it is recommended that future research on cannabinoids incorporate gender stratification to rigorously assess their impact [33].

## 2. Cannabinoids: Types and Mechanisms of Action

### 2.1. Phytocannabinoids

The historical and medicinal relevance of Cannabis sativa is rich, encompassing both therapeutic and recreational applications. With over 120 C21 terpenophenolic compounds known as phytocannabinoids, Cannabis sativa is a prime source of bioactive natural compounds. THC, discovered in 1971, remains dominant among these, and its discovery led to the identification of the ECS, which comprises CB1 and CB2 receptors. Despite its psychotropic effects limiting its medical utility, THC, along with other phytocannabinoids, holds promise for treating conditions like pain, anxiety, and cachexia. Contemporary research is exploring the biosynthesis of phytocannabinoids in various species including Cannabis, Rhododendron, and Radula, as well as the potential for engineering cannabinoids with enhanced properties via synthetic biology strategies [34,35].

Similar bioactive constituents, namely phytocannabinoids, are also present in hashish and marijuana, both of which are derivatives of Cannabis sativa L. Traditional pharmacology focused primarily on these compounds interacting with CB1 and CB2 receptors. However, newer insights suggest a more complex interaction profile involving multiple targets. The molecular pharmacology of key phytocannabinoids, particularly THC and CBD, is a focal point in understanding their diverse range of actions [36,37].

Intricacies in the composition of phytocannabinoids involve a range of pathways and variations in side-chain composition as well as the degree of isoprenyl residue oligomerization. The complexity of these compounds extends to their varying origins, which include not just higher plants but also liverworts and fungi. Factors like heat, light, and atmospheric oxygen can induce non-enzymatic alterations in these compounds, affecting key constituents like CBG, CBD, THC, and CBC. Not confined to CBRs, these bioactive molecules engage with a variety of targets, such as thermo-TRPs and transcription factors like PPARs, suggesting their potential as an investigational class of drugs with actions beyond the ECS [38,39,40].

The limitations of conventional antiepileptic medications, characterized by suboptimal efficacy and adverse side effects, render the exploration of alternative therapies imperative. Phytocannabinoids, notably THC and CBD, offer a promising avenue in this context as they have exhibited anticonvulsant properties with comparatively fewer adverse effects in both preclinical and initial human studies. With the growing global acceptance of cannabis-derived products as medical interventions, an understanding of their neurochemical mechanisms of action is essential. THC functions as a partial agonist at the cannabinoid 1 and 2 receptors (CB1/2), leading to typical outcomes such as euphoria and relaxation. However, it may also induce dysphoria, anxiety, and manifestations of psychosis in certain cases [41].

CBD and its propyl analog, cannabidivarin (CBDV), have been the focus of increasing scientific inquiry due to their wide spectrum of therapeutic attributes, including anti-inflammatory, anti-nausea, anti-tumor, anti-convulsant, anxiolytic, and neuroprotective qualities. Despite the plethora of molecular targets with which phytocannabinoids interact across various body systems, a comprehensive understanding of their mechanisms of action remains elusive. The nematode C. elegans serves as a vital model organism in this context, sharing approximately 60% of the genes associated with human pathologies and exhibiting remarkable neural circuitry and G-protein-coupled receptor (GPCR) signaling similarities to mammals [42].

With respect to major depressive disorder, compelling evidence suggests that the activation of CB1 receptors may function as a protective mechanism in humans, either directly or indirectly. This proposition is further supported by the negative mood effects, including depression and suicidal thoughts, observed in obese patients treated with CB1 antagonists. Moreover, the silencing of CB1 receptors in specific neural circuits has been shown to elevate susceptibility to stress, potentially triggering a cascade of stress-related disorders, including depression [43,44].

Despite the long-standing recognition of the therapeutic potential of phytocannabinoids and their antioxidative capabilities, the operational mechanisms remain under-explored. Recent investigations have employed density functional theory (DFT) calculations to scrutinize the radical scavenging abilities of CBD and cannabidiolic acid (CBDA). These studies highlight the effectiveness of these compounds in neutralizing hydroperoxyl radicals in polar physiological environments, albeit with diminished efficacy in lipid-rich media. This focus has also extended to investigating the antiradical properties of eight key compounds from all the primary families of phytocannabinoids, including cannabinol (CBN), THC, cannabichromene (CBC), cannabicyclol (CBL), cannabielsoin (CBE), CBD, cannabifuran (CBF), and cannabigerol (CBG) [45].

The intricate interactions between the CBRs CB1 and CB2 have revealed a fascinating phenomenon: the formation of complex molecular assemblies known as CB1/2RHet complexes. These complexes are noteworthy for their potential to modulate CB1R-mediated effects. Investigations have been extended into the impact of various cannabinoid compounds on the formation of these receptor complexes [46]. A model utilizing HEK-293T cells was employed for this purpose; these cells were subjected to transfection with a consistent quantity of CB1R-RLuc cDNA while varying amounts of CB2R-GFP2 cDNA were also introduced. The resulting data were plotted onto a saturable bioluminescence resonance energy transfer (BRET) curve. Key observations from this experiment include the BRETmax value, determined to be 214 ± 15, and the BRET50 values, identified as 48 ± 9, both of which suggest targeted and specific interactions between CB1R and CB2R [47].

### 2.2. Endocannabinoids

The isolation of a cannabinoid receptor ligand, arachidonylethanolamide or anandamide, from porcine brain tissue has provided significant insights into its physiological effects. Anandamide effectively interferes with radiolabeled cannabinoid ligand binding to synaptosomal membranes from rat brains. Moreover, the compound exhibits dose-dependent inhibitory actions in electrically stimulated mouse vas deferens [48]. Further investigations reveal that anandamide acts as a cannabinoid agonist, effectively inhibiting the forskolin-induced activation of adenylate cyclase in N18TG2 cells, similar to the effects of HU-210, while (+)-HU-211 demonstrates minimal impact [1]. The multifaceted actions of anandamide encompass the blockage of voltage-gated calcium channels, the activation of inwardly rectifying potassium currents, G-protein binding, and the induction of multiple cellular signaling pathways, among other effects [49,50].

2-Arachidonoylglycerol (2-AG), a molecular variant of monoacylglycerol, bears a structural resemblance to anandamide. It is unique in attaching arachidonic acid at the second position of its glycerol framework. Studies indicate that 2-AG binds to CBRs on synaptosomal membranes derived from rat brain tissue, albeit with reduced potency compared to anandamide [51]. This raises the possibility that arachidonic-acid-containing monoacylglycerols could function as endogenous ligands for CBRs in specific neural contexts. Unlike conventional neurotransmitters, which act through vesicular secretion from synaptic terminals, anandamide and 2-AG may be synthesized on demand via the stimulation-triggered cleavage of separate phospholipid precursors located within neuronal membranes [52].

Research indicates the presence of cannabinergic modulation within the basal ganglia, as evidenced by the effects of the CB1 receptor antagonist SR141716 [53]. This antagonist led to increased locomotion in mice and induced stereotypies in rats. The activation of CBRs has also been found to significantly reduce electrically-induced dopamine release in rat striatal slices and to potentiate the symptoms of neuroleptic-induced catalepsy [54]. Importantly, blocking the CBRs effectively removed the inhibitory regulation mediated by endogenously released anandamide, thereby amplifying quinpirole-induced motor activation [55]. The absence of any observable effects of SR141716A when administered alone, at doses similar to those used to augment quinpirole-induced motor activation, suggests that anandamide’s behavioral effects may be dependent on D2 receptor stimulation, potentially countering dopamine-D2-facilitated psychomotor activity [56].

Anandamide has been shown to activate transient receptor potential (TRP) channels, specifically TRPV1, under certain conditions. While both CBRs and TRP channels seem to contribute to its effects, their individual roles appear to be variable [57]. Anandamide is also known to activate alpha and gamma peroxisome proliferator-activated receptors (PPARs), which have a substantial impact on gene transcription. The inhibition of their degradation via FAAH increases the levels of anandamide as well as other N-acylamides that modulate PPARα receptors [58,59].

Both THC and anandamide are classified as low efficacy agonists. Under specific conditions, such as low receptor density or limited post-receptor effectors, these compounds may function as antagonists by negating the CB1 receptor signaling initiated by 2-AG [60].

The ECS is a complex network that includes endogenous cannabinoids (eCBs) like N-arachidonoylethanolamine (anandamide or AEA) and 2-arachidonoylglycerol (2-AG), biosynthetic enzymes such as NAPE-specific phospholipase D and Diacylglycerol lipase- α, and degradative enzymes like fatty acid amide hydrolase (FAAH) and monoacylglycerol lipase (MAGL) [61,62,63,64]. The receptors for these substances, termed cannabinoid receptors (CBRs), also form integral components of this system. Notably, these eCBs interact not just with the primary CBR subtypes (CB1R and CB2R) but also with various other receptors, including transient receptor potential vanilloid type 1 (TRPV1) cation channels, GTP-binding protein-coupled receptor GPR55, abnormal-CBD receptor, and peroxisome proliferator-activated receptors (PPARs) [65,66].

eCBs serve as critical regulators of synaptic transmission through various physiological feedback mechanisms designed to counteract either the overexcitation or inhibition of synapses. These mechanisms include retrograde signaling, which leads to the depolarization-induced suppression of inhibition (DSI) at GABAergic synapses and the depolarization-induced suppression of excitation (DSE) at glutamatergic synapses. The presynaptic location of CB1R allows eCBs to influence other neurotransmitters, such as opioid peptides, acetylcholine, and 5-hydroxytryptamine (5-HT), even if CB1Rs may not be expressed in nigrostriatal dopaminergic neurons [67,68]. Nonetheless, the functionality of these neurons can be profoundly affected either by the activation or blockade of ECS components present in nearby neuronal subpopulations like GABAergic, glutamatergic, and opioidergic neurons that interconnect with dopaminergic neurons. Moreover, dopaminergic neurons can produce extracellular peptide-binding proteins, enhancing retrograde signaling at both excitatory and inhibitory synapses [69,70].

Additional theories suggest that eCBs modulate dopamine (DA) transmission through interactions with TRPV1 receptors and the formation of heteromers with metabotropic receptors such as dopamine D1 and D2 receptors. The presence of CB2R also implies a direct role of eCBs in modulating dopamine transmission, thus expanding the scope of their physiological and potentially therapeutic roles [71,72].

### 2.3. Synthetic Cannabinoids

Phytocannabinoids, predominantly comprising THC, originate from plant sources such as cannabis. On the other hand, synthetic cannabinoids present in products like Spice include various compounds such as naphthoylindoles, benzoylindones, and phenylacetylindoles [73]. Notably, the composition of synthetic cannabinoids can vary substantially across different Spice products and even within the same batch or package (European Monitoring Centre for Drugs and Drug Addiction 2009). Beyond synthetic cannabinoids, Spice formulations may include a range of other substances such as additives, preservatives, fatty acids, amides, esters, and additional psychoactive compounds like the benzodiazepine phenazepam and an active metabolite of tramadol. Some formulations have even been found to contain Salvia divinorum, Kratom, or cannabis, although the impact of these additional substances on the overall effects of Spice remains unclear [74].

Neurologically, Spice has been associated with a diverse array of symptoms that include tremors, ataxia, nystagmus, fasciculations, and hypertonicity, as well as hyperflexion and hyperextension. Cognitive impairments affecting attention, concentration, and memory have also been reported, along with a compromised ability to operate machinery. Palpitations frequently accompany feelings of panic, complicating efforts to discern whether these symptoms stem from underlying anxiety. Even after the acute phase of palpitations subsides, residual irregularities may persist. Additional observed symptoms include xerostomia (commonly known as “cotton mouth”), reddened conjunctiva, changes in pupil size leading to either constriction (miosis) or dilation (mydriasis), heightened sensitivity to light, and persistent coughing and inflammation or injury to the lungs [73].

### 2.4. Cannabinoid Receptors and Signaling Pathways

CB-1R receptors are ubiquitously distributed throughout the nervous system, with pronounced concentrations in regions such as the hippocampus, association cortex, cerebellum, and basal ganglia, among others [75,76]. In contrast, CB-2R receptors are principally localized in gastrointestinal and lymphatic tissues, as well as specific CNS locations such as the dorsal nucleus of the vagus nerve, spinal trigeminal nuclei and nucleus ambiguous [76,77]. Both CB-1R and CB-2R function as G-protein-coupled receptors, modulating the release of neurotransmitters like glutamate, dopamine, and acetylcholine through the inhibition of adenyl cyclase activity via G0/Gi proteins. Additional neurotransmitter pathways, including serotonergic, GABAergic, and NMDA (N-methyl-D-aspartate), are indirectly modulated. A noteworthy observation pertains to the TRPV1 receptor, which has been identified in the basal ganglia through advanced imaging techniques [78]. The ECS employs feedback mechanisms to regulate synaptic transmission, affecting cell development, differentiation, and apoptosis via the MAPK/ERK pathway [79]. 

Recent studies have classified CBRs in both rats and humans as members of the seven-transmembrane GTP-binding protein-coupled receptor family. Investigations demonstrated that exogenous cannabinoids could suppress forskolin or secretin-induced adenylate cyclase activity and inhibit the opening of the N-type calcium channel, processes that are rendered ineffectual when pretreated with pertussis toxin (PTX), thereby implicating the G1/G0 GTP-binding proteins in these signaling pathways [80].

Furthermore, the CB1 receptor has been identified to contain 472 amino acids in humans and 473 in rats [81]. A second cannabinoid receptor, CB2, was also successfully isolated and was found to have a similar architecture consisting of 360 amino acids [82]. Despite sharing only 44% overall similarity, both receptors share 68% resemblance in their transmembrane domains and are coupled to Gi/Go proteins. Various compounds, including those with antagonist or inverse-agonist properties such as SR141716A for CB1 and SR144528 for CB2, have been developed to interact with these receptors [83]. It is posited that CB1 is fundamentally implicated in the regulation of cognition, memory, and motor activities [84].

## 3. The Pharmacology of Cannabinoids

### 3.1. Absorption, Distribution, Metabolism, and Excretion (ADME)

In the realm of drug discovery, a comprehensive selection of target and ligand molecules from cyano-bacterial species was carried out based on their biological and pharmacological attributes. These selections were further refined through homology modeling, molecular docking, and molecular dynamics (MD) simulations. A highlight was the utilization of an in silico tool, Maestro v10.2’s Quikprop, for the assessment of their absorption, distribution, metabolism, and excretion (ADME) properties. This computational approach adhered to well-established guidelines such as the Rule of Five and considered both physicochemical parameters and toxicology measures. The insights gained facilitated the accurate prediction of pharmacokinetic properties, corroborating the vital role of in silico methods in drug discovery processes [85].

In evaluating the pharmacodynamic, pharmacokinetic, and toxicity profiles of the selected cannabinoids, ADME/TOPKAT prediction proved to be highly instructive. The compounds exhibited varied levels of human intestinal absorption, blood–brain barrier penetration, and solubility. Furthermore, the compounds also displayed varying degrees of plasma protein binding and hepatotoxicity. These analyses collectively contributed to a nuanced understanding of the complex pharmacokinetic and pharmacodynamic profiles of the studied cannabinoid compounds [86].

The integral role of ADME studies in both drug discovery and development is well-documented. These studies not only predict human pharmacokinetic properties but also help establish correlations with pharmacodynamic assessments in commonly employed animal models for nonclinical investigations. The employment of specialized methodologies, such as [14C]-S-777469, has proven to be invaluable for the in-depth analysis of specific agonists like S-777469, offering nuanced insights into their behavior within human systems and broadening the understanding of their pharmacological profiles [87].

In the context of pharmacology, it has been observed that serious illness substantially alters all facets of drug disposition, including absorption, distribution, metabolism, and excretion (ADME). Such alterations manifest in multiple dimensions, from disrupted oral absorption and bioavailability to shifts in drug distribution patterns and metabolic pathways. These modifications suggest that pharmacokinetic models based on healthy subjects may not be wholly applicable to those with illnesses, thus necessitating more nuanced approaches for this demographic [88].

In studies focusing on CBRs, the compounds were stringently assessed for their binding affinities to human and mouse CB2 receptors (CB2Rs), as well as their selectivity towards human CB1 receptors (CB1Rs). Among the tested compounds, ”Compound 2f” stood out for its strong affinity and selectivity for CB2R. This compound was further subjected to advanced metabolic pathway analyses, including incubation with human and rat liver microsomes. Additional in vivo tests were conducted to evaluate the metabolic stability of [18F]2f, thereby adding to the compound’s potential candidacy for PET tracer applications [89].

Regarding the neuroprotective potential of cannabinoids, extensive analyses were carried out on three specific receptors (CB1, CB2, and CB3) and selected phytocannabinoids, including THC and CBD, as well as endogenous cannabinoids like anandamide (AEA) and 2-arachidonoylglycerol (2-AG). These studies shed light on the intricate interplay between cannabinoids and their molecular targets. Furthermore, the ADME profiles of these compounds indicate favorable drug-like characteristics, thereby supporting their potential applicability in the treatment of neurodegenerative or other neurological conditions [90].

The foundational premise for the detection of drugs in sweat is anchored in the pharmacokinetic understanding of a drug’s absorption, distribution, metabolism, and excretion (ADME) cycle. During this cycle, a fraction of the drug is anticipated to be excreted through sweat. The presence of lipophilic compounds in the bloodstream is modulated by factors such as their pKa (acid dissociation constant) values and the pH of the fluids they enter. Employing a modified Henderson–Hasselbach equation, which incorporates both pKa and pH, allows for the theoretical determination of the fluid/plasma concentration ratio (F/P ratio). Passive diffusion, which is governed by concentration gradients, typically enables drugs to permeate sweat, with only the unbound fractions making this transition. Additionally, the more acidic pH of sweat compared to blood creates a tendency for basic drugs to accumulate within its layers [91].

As for the pharmacokinetics of inhaled cannabinoids, inhalation and intravenous administration yield similar profiles. Post inhalation, peak plasma concentrations of THC and CBD are rapidly achieved, typically within 3–10 min. The bioavailability of inhaled THC is estimated to range between 10 and 35%, a value influenced by various factors including inhalation patterns, breath-holding durations, and the specifications of the inhalation device used. CBD, when inhaled, exhibits an average systemic bioavailability of approximately 31%, presenting a plasma concentration–time profile analogous to that of THC [92,93,94].

### 3.2. Pharmacokinetics and Pharmacodynamics

The administration of natural cannabis products and cannabinoids predominantly occurs via inhalation or oral consumption, whereas methods like rectal administration, sublingual intake, transdermal application, eye drops, or aerosols are generally considered to have limited practical utility. The pharmacokinetics of THC in particular are strongly influenced by the mode of administration. For instance, inhalation leads to a swift increase in plasma concentration and psychotropic effects manifest within seconds to minutes, reaching a peak within 15–30 min and dissipating over a span of two to three hours. Contrastingly, oral administration yields delayed psychotropic effects that appear 30 to 90 min post consumption, peaking between 2 and 3 h and lasting 4–12 h, contingent on the dosage and specific effects [95].

The pharmacokinetics of novel psychoactive substances (NPSs) serve as a crucial framework for understanding organismal responses to drug administration. While forensic casework offers data on cannabinoid concentrations in human users, it offers restricted insights into the pharmacokinetics of individual samples. Preclinical research utilizing laboratory animals offers more comprehensive data concerning the biological effects of synthetic cannabinoids, although it often surfaces after these substances have exited market circulation. In light of these considerations, the pharmacokinetics and pharmacodynamics of 5F-MDMB-PICA, an FDA-approved synthetic cannabinoid that is popular in the USA, have been characterized. A validated analytical methodology has been established that is capable of quantifying 5F-MDMB-PICA and its primary metabolites in rat plasma. Previous studies have identified 12 and 22 metabolites of 5F-MDMB-PICA in vitro [96].

Although CBD is often presumed to mitigate some of the undesirable side effects of THC, such as its anxiety-inducing properties, controlled clinical investigations have yielded inconsistent conclusions. Some studies indicate that CBD can attenuate specific acute effects of THC, while others suggest that CBD may augment THC’s pharmacodynamics, resulting in more profound drug effects. Yet other findings imply that CBD might not alter either the pharmacodynamics or pharmacokinetics of THC [97].

The distribution of cannabinoids within bodily tissues is significantly influenced by their lipophilic nature. THC, for instance, exhibits a substantial distribution volume (ranging from 5.7–10 L/kg) that is attributable to its lipophilic properties. Similarly, CBD’s distribution volume is notable, allowing for its efficient penetration into the brain, adipose tissue, and various organs. The chronic consumption of cannabinoids tends to result in gradual tissue accumulation, further amplifying their distribution volume. The metabolism of cannabinoids is primarily hepatic, although extra-hepatic metabolism also occurs in other organs such as the brain, intestines, and lungs. Cytochrome P450 (CYP 450) enzymes, which are predominantly found in liver tissue, play a vital role in the metabolic breakdown of THC into its main components through decarboxylation, epoxidation, and oxidation processes, leading to D11-hydroxy-THC (D11-OH-THC) and D11-carboxy-THC (D11-COOH-THC). Tissues expressing CYP 450 enzymes also contribute to the extra-hepatic metabolism of THC [98,99].

### 3.3. Factors Influencing Cannabinoid Effects

The question of marijuana serving as a gateway drug has been the subject of extensive inquiry. A particular computational model replicates observed phenomena frequently cited to substantiate the gateway effect. However, the model does not indicate a direct causal relationship between marijuana use and the initiation of using hard drugs. Another facet of the argument focuses on the relative risk associated with the user’s age at the time of the initiation of marijuana use. This variant of relative risk associates the initiation of hard drug use with user characteristics like age rather than solely marijuana use and thus fails to provide compelling evidence for the existence of a gateway effect [100].

Legal frameworks and public perception play pivotal roles in the variations in the availability of synthetic cannabis plant material. Originating primarily in China, these new psychoactive substances (NPSs) are influenced by law enforcement efforts, media coverage, and legislative changes in both their country of origin and in destination countries, affecting their global availability [101].

Maternal marijuana usage during gestation has been scrutinized for its potential impact on the neurobehavioral development of offspring. Animal models reveal enduring negative consequences associated with cannabis exposure during gestation and lactation, particularly with the rise of cannabis use among adolescents. The long-term administration of cannabinoid agonists in the periadolescent phase in animals has been correlated with enduring behavioral changes and increased susceptibility to conditions like psychosis or other neuropsychiatric disorders [102].

Research involving heavy adolescent users of cannabis suggests prolonged deficits in learning and working memory which endure up to six weeks post cessation. These findings are particularly concerning given the ongoing process of neuromaturation during adolescence. Rodent models corroborate this, showing more pronounced memory impairments in animals exposed to cannabinoids during adolescence as opposed to later exposure. Moreover, adult humans who initiated cannabis use in their adolescent years experience greater cognitive dysfunction compared to those who initiated it later. This adds credence to the hypothesis that adolescents might be more susceptible to the neurocognitive disruptions associated with chronic and heavy marijuana usage, although the role of preexisting risk factors remains an area for further investigation [103].

Scholarly investigations into the potential impact of permissive state medical marijuana laws (MMLs) on recreational cannabis use have yielded inconclusive results. One underexplored avenue is the effect of MMLs on the average potency of consumed marijuana. It is theorized that heightened potency could indirectly influence individual consumption patterns as less material would be needed to achieve intoxication, potentially reducing overall usage. This line of inquiry, while theoretically compelling, has yet to gain substantial attention in academic circles [104]. Cannabinoids, initially synthesized in acidic forms such as THC, CBD, CBN, CBG, CBC, and CBND, have showcased considerable therapeutic promise. Their documented benefits range from alleviating nausea in chemotherapy patients and enhancing appetite in HIV-positive individuals to reducing spasticity in adults with multiple sclerosis. Additional potential applications include antitumor effects and the management of conditions like glaucoma, epilepsy, and schizophrenia [105]. An understanding of both the pharmacokinetic and pharmacodynamic properties of cannabinoids is essential to fully appreciating their biological impacts. While pharmacodynamic studies have confirmed anti-inflammatory, antiviral, and anticancer properties, pharmacokinetic attributes can vary considerably among individuals. Numerous factors, including prior consumption habits, pharmacogenetics, body size, health status, diet, and microbiome composition, along with dosage and the route of administration, influence the cannabinoids’ pharmacokinetic profiles. Empirical research employing subjective self-reports, cognitive task assessments, and neurophysiological evaluations like electroencephalography (EEG) and event-related potentials (ERPs) has elucidated some effects of THC consumption. Compared to placebo conditions, THC-infused cigarettes were associated with expected shifts in mood, behavior, and brain activity. These alterations included diminished task performance and attenuated EEG power and ERP components linked to attentional processes during memory-intensive tasks. Importantly, these effects largely lacked dose dependence. Furthermore, variations in the concentrations of other cannabinoids like CBC and CBD did not significantly influence these outcomes, underscoring the primary bioactive role of THC and its metabolites and affirming the utility of EEG/ERP as biomarkers of its impact [106].

## 4. Cannabinoids and Neurodegenerative Diseases

### 4.1. Alzheimer’s Disease

In the context of Alzheimer’s disease (AD), in which existing therapies offer limited efficacy, research is gradually turning toward the endogenous cannabinoid system as a promising therapeutic target. This system comprises CB1 and CB2 receptors, intrinsic ligands, and enzymes for synthesizing and degrading eCBs. Experimental models of Alzheimer’s have demonstrated the potential of activating CB1 and CB2 receptors with non-psychoactive agonists to produce favorable outcomes. These include attenuating the deleterious effects of beta-amyloid peptides and tau phosphorylation while promoting brain repair mechanisms. Although much of this evidence is derived from animal models simulating the pathology of AD, preliminary clinical data supports the role of cannabinoids in ameliorating the behavioral symptoms associated with Alzheimer’s disease, particularly when using THC analogs such as nabilone or DRO. Notably, adverse effects like euphoria, somnolence, and fatigue were generally manageable and did not necessitate the cessation of treatment [107,108].

Moreover, cannabinoids possess neuroprotective properties that could be crucial in treating AD. For instance, they can diminish tau phosphorylation and mitigate the negative impact of beta-amyloid-induced oxidative stress while promoting neurotrophin expression and neurogenesis. THC itself shows the capacity to inhibit acetylcholinesterase activity, potentially slowing the progression of the disease [109,110]. CBRs on microglial cells also present a unique intervention point for mitigating AD-associated neuroinflammation without inducing psychoactive effects [111,112]. Epidemiological data further corroborate ECS’s role in AD, especially as nonsteroidal anti-inflammatory drugs (NSAIDs) have been observed to reduce risk. The findings suggest that eCBs may offer a protective mechanism against beta-amyloid-induced damage. Indeed, recent studies indicate that inhibiting endocannabinoid uptake could reverse beta-amyloid-induced neurotoxicity and cognitive impairment, emphasizing the therapeutic potential of augmenting endocannabinoid levels in the brain [113,114,115].

Despite skepticism around the psychoactive properties of cannabis, accumulating evidence suggests that THC, CBD, and synthetic analogs hold therapeutic potential in ameliorating memory impairment associated with AD. Specifically, these substances have displayed consistent efficacy in rodent and human studies, validating the CB1 receptor as a candidate for therapeutic targeting. Furthermore, the anti-inflammatory capabilities of cannabis and THC align with the need to mitigate neuroinflammation in neurodegenerative diseases like AD. To harness the therapeutic utility of THC effectively, it is crucial to discern its medical attributes from its recreational effects, as supported by preclinical and clinical findings [116].

Additional research has illuminated the chronic administration of THC and CBD as promising in mitigating memory impairments during the advanced stages of the pathology of AD, as demonstrated in APP/PS1 mice. Interestingly, these compounds failed to exert similar benefits during the early stages of the disease, leaving Ab deposition and gliosis unaltered. Therapeutic outcomes in aged APP/PS1 mice correlated with improved synaptic function, which was characterized by specific changes in metabotropic glutamate receptor 2/3 and GABA-A Ra1 levels. Both CB1 receptor agonists and THC have been shown to induce the release of brain-derived neurotrophic factor (BDNF), implying that this mechanism could be central to THC’s neuroprotective properties. Given BDNF’s role in regulating synaptic plasticity, its upregulation could offer a therapeutic pathway to restore synaptic function in AD patients. However, it should be noted that any therapeutic endeavor involving cannabinoids must consider the long preclinical phase of AD and the necessity for early diagnosis for therapies to be effective [117,118,119,120].

The therapeutic capacity of THC in targeting intraneuronal amyloid-beta (Ab) has been found to be promising, although its psychoactive attributes pose social challenges. A recent study sought to examine a selection of non-psychoactive cannabinoids for their neuroprotective qualities. Utilizing a well-regarded Alzheimer’s disease drug discovery platform, these compounds underwent thorough assessments via assays that evaluate toxicities pertinent to the aging brain, such as proteotoxicity, trophic support loss, energy depletion, and oxidative stress. The study also scrutinized the cannabinoids’ effects on microglial inflammation. Preliminary results indicate that many of these cannabinoids manifest significant neuroprotective properties across a range of assessments, suggesting that they are viable candidates for clinical applications in treating neurodegenerative disorders [121].

The potential therapeutic roles of cannabinoids in addressing late-onset Alzheimer’s disease (LOAD) and other prevalent conditions among the elderly have increasingly captured scholarly interest. A host of in vitro and in vivo investigations corroborate the capacity of cannabinoids to mitigate oxidative stress, neuroinflammation, and the formation of hallmark LOAD markers like amyloid plaques and neurofibrillary tangles. Additionally, population-based studies suggest that cannabinoids may ameliorate symptoms commonly associated with dementia, such as behavioral disturbances. This comprehensive review elaborates on the burgeoning body of evidence suggesting cannabinoids’ potential utility in treating LOAD while also offering critical insights into their efficacy, safety, and pharmacokinetics when administered as treatment in dementia-afflicted populations [122].

### 4.2. Parkinson’s Disease

The burgeoning interest in cannabinoid treatment for alleviating Parkinsonian symptoms, such as dyskinesia and tremors, has been significantly propelled by media coverage and anecdotal evidence disseminated via online platforms. Carroll et al. executed a rigorous randomized, double-blind, placebo-controlled crossover trial utilizing a standardized whole-plant extract with a specific THC concentration and a THC:CBD ratio of approximately 2:1, with the dosage tailored to individual body weight. Despite the double-blind design, a majority (71%) of the 17 participating Parkinson’s disease patients were able to correctly identify their treatment arm. The study’s primary findings indicated that the oral cannabis extract was well-tolerated but yielded no notable changes in Parkinsonian symptoms. Key outcome measures, such as the Unified Parkinson’s Disease Rating Scale (UPDRS) and Rush Dyskinesia Rating Scale, along with secondary outcomes like pain scores and sleep quality assessments, failed to evince any significant treatment effects on levodopa-induced dyskinesia (LID) [123].

In the realm of neuroprotection for Parkinson’s disease (PD), cannabinoids exhibit considerable promise, particularly in mitigating factors like excitotoxicity, calcium influx, glial activation, and oxidative damage—all of which are implicated in the progressive degeneration of nigral neurons [124,125]. Although preclinical evidence is robust in supporting the neuroprotective potential of cannabinoids, clinical investigations remain markedly limited. Despite the pressing necessity for innovative therapeutic approaches that extend beyond dopaminergic replacement therapy and the lack of existing effective neuroprotective strategies, the clinical exploration of cannabinoid-based treatments has been constrained. This dearth of clinical studies persists despite the compelling preclinical data, underscoring the urgent need for further research to bridge the gap between preclinical promise and clinical applicability in the treatment of PD [126].

The ECS has been highlighted as a key player in the functioning of the basal ganglia, which is critically implicated in movement disorders such as Parkinson’s disease (PD). While preclinical studies suggest that the modulation of cannabinoid (CB) signaling may alleviate motor symptoms—including levodopa-induced dyskinesias (LIDs)—the clinical translation of these findings remains insufficiently explored [127,128]. LIDs often manifest as a result of repetitive dopamine receptor stimulation, leading to a heightened sensitivity in CB receptor-related striatal signaling. Despite advancements in understanding the molecular interplay between cannabinoids (CBs) and dopamine (DA), their applicability in treating PD and LIDs is still an area that demands further investigation [123,129,130]. 

A limited number of clinical studies exist, such as a Class III randomized, double-blind, placebo-controlled crossover trial exploring the effect of nabilone, a CB1 and CB2 agonist. The trial demonstrated a reduction in both Rush Dyskinesia Disability Scale scores and the total LID time, suggesting the drug’s potential therapeutic efficacy. In the context of PD, levodopa-induced dyskinesia (LID) has been shown to involve hyperactivity in the lateral segment of the globus pallidus (GPl). The activation of CBRs in this region can modulate the reuptake of GABA and potentially enhance neurotransmission, which could theoretically ameliorate the symptoms of dyskinesia. Supporting this hypothesis, a controlled clinical study—a randomized, double-blind, placebo-controlled crossover trial involving seven PD patients—provided empirical evidence that the cannabinoid receptor agonist nabilone significantly reduced occurrences of LID [66]. Despite the insights gained from preclinical research, definitive clinical evidence regarding the therapeutic efficacy of cannabinoid therapies for PD and associated LIDs is still sparse, warranting further in-depth clinical evaluations [131].

In a study employing a rat model with 6-hydroxydopamine-induced lesions, the post-lesion-onset administration of THC led to a resurgence of neuronal injury two weeks after the cessation of the cannabinoid treatment. This finding prompts questions concerning the nature of THC’s protective effects against 6-hydroxydopamine toxicity—whether they are inherently neuroprotective and sustained post treatment or merely transient upregulatory responses. Concurrent investigations probed alterations in the efficacy of CB1 receptors within the caudate putamen and substantia nigra two weeks post toxin administration. Emerging research substantiates that the prolonged hyperactivation of these receptors parallels observations in other Parkinson’s disease models. Importantly, the chronic administration of THC appeared to significantly attenuate dopaminergic neuronal injury in hemiparkinsonian rats, corroborating previous evidence of the neuroprotective properties of cannabinoids—whether plant-derived, synthetic, or endogenous—across various in vivo and in vitro models of neuronal damage [132].

The collective body of evidence suggests a role for CB2 receptor activation in mitigating inflammation and neuronal degeneration within a 6-hydroxydopamine (6-OHDA) model of Parkinson’s disease (PD). An investigation into the impacts of cannabinoids on neuronal survival post 6-OHDA exposure highlighted microglia-mediated effects. CB receptor agonists such as 8-THC and 9-THC have been demonstrated to suppress the release of proinflammatory cytokines, including tumor necrosis factor-alpha (TNF-α) and interleukins, from human monocytes [133]. Additional compounds, such as WIN-55,212-2, CBD, and JWH-133 (a selective CB2 receptor agonist), have been reported to counteract ATP-induced increases in intracellular calcium concentrations in N13 microglial cell lines. The effects of JWH-133 and WIN-55,212-2 were completely nullified by the CB2 antagonist SR 144528, underscoring their CB2-receptor dependency. Interestingly, such antagonistic effects were absent in CBD-treated cells, indicating the existence of CB2-independent mechanisms that potentially contribute to the observed neuroprotective effects [134].

### 4.3. Huntington’s Disease

Since the 1980s, there has been a remarkable surge in the field of cannabinoid pharmacology, culminating in the development of innovative cannabinoid-based pharmaceuticals to address an array of medical conditions. Notable medications that emerged during this period include Cesamet (nabilone) and Marinol, which received approval for the treatment of chemotherapy-induced nausea and vomiting in oncology patients and anorexia–cachexia in the context of AIDS therapy, respectively. The most recent addition to this burgeoning domain is Sativex, an oromucosal spray developed by GW Pharmaceuticals Plc, comprising equimolar concentrations of THC and CBD for optimal efficacy. Beyond MS, the multifaceted pharmacological profile of Sativex^®^—including its demonstrated analgesic, antitumoral, anti-inflammatory, and neuroprotective properties in preclinical settings—has catalyzed ongoing research into its applicability for treating additional neurological disorders [135,136].

One neurodegenerative condition that has garnered particular attention in this context is Huntington’s disease (HD). HD is an autosomal-dominant disorder characterized by the presence of excessive CAG repeats in one allele, leading to polyglutamine (polyQ) expansion in the huntingtin protein. The disease predominantly affects striatal and cortical neurons and may manifest clinically as chorea or dementia. In an investigation into the role of CB2 receptors in excitotoxicity-induced striatal neurodegeneration, the anti-inflammatory compound minocycline was administered to CB2 receptor-deficient mice. The study found a significant reduction in excitotoxicity-induced seizures and enhanced motor coordination, and balance, as indicated by performance on a RotaRod test. Moreover, minocycline alleviated glial activation and decreased the loss of medium-sized spiny neurons in these CB2-receptor knockout mice. These results underscore the significance of CB2 receptors in mediating microglia-driven neuroinflammatory processes and suggest that the efficacy of HU-308, a CB2 receptor agonist, may rely substantially on the modulation of microglial activation [137].

In the realm of Huntington’s disease (HD), a particular focus has been placed on the potential use of cannabinoids in treating dystonia, a frequent motor symptom. One study revealed that a cohort of early-onset HD patients experienced a notable amelioration of the symptoms of dystonia upon the initiation of cannabinoid treatment, affirming the therapeutic promise of cannabinoids in mitigating motor dysfunctions, especially dystonia, in early-onset HD cases [138].

An additional study sought to elucidate the neuroprotective capabilities of cannabinoids within the context of neurodegenerative disorders, using an experimental model that mimicked the mitochondrial complex II deficiency frequently observed in HD. Here, the effects of THC, a nonselective cannabinoid receptor agonist, and SR141716, a specific CB1 receptor antagonist, were examined during malonate-induced striatal toxicity. Contrary to expectations, both THC and SR141716 exacerbated malonate-induced lesions. These findings indicate the complexity of manipulating the ECS for neuroprotection in HD and suggest that targeting highly selective CB1 receptor agonists may be necessary to effectively mitigate neurodegeneration [132]. 

In a longitudinal study that employed a R6/1 mouse model to investigate the effects of various cannabinoid treatments on Huntington’s disease (HD), several key observations were made over a 20-week period. Although changes in female body weight were statistically insignificant and consequently omitted from the report, marked disparities were noted in male cohorts. Specifically, wild-type mice exhibited a consistent pattern of weight gain, whereas in R6/1 mice, weight gain reached a plateau after the twelfth week. Various eight-week treatment regimens with HU210, THC, or URB597, initiated at the 12-week mark, failed to mitigate behavioral deficits in the R6/1 mice. However, molecular assays indicated that URB597 effectively preserved CB1 receptors in the striatum, while HU210 resulted in an aggregation of ubiquitin-positive protein [139] (Table 1).

The study challenges the extant literature on the subject by revealing no significant downregulation of CB1 receptors due to chronic drug therapy. This discrepancy may be attributable to variations in treatment durations and the specific brain regions analyzed. Moreover, the study underscores the unexpected increase in seizure events in the R6/1 mice following HU210 treatment, thereby raising questions regarding the safety and appropriateness of utilizing highly potent cannabinoid agonists in HD therapy (Figure 1).

Collectively, these observations offer nuanced insights into the intricate relationship between cannabinoids and HD, thereby accentuating the need for further research. The study highlights the necessity to clarify the role of CB1 receptors in disease progression and assess the viability of targeted therapeutic interventions [139] (Table 2).

### 4.4. Multiple Sclerosis

The potential therapeutic efficacy of cannabinoids in treating the symptoms associated with multiple sclerosis (MS) and spinal cord injury has gained some empirical support, primarily from a limited set of eight clinical trials and a solitary case study focused on spinal cord injury. Among these, five studies concentrated on the administration of oral THC, offering preliminary evidence of its beneficial effects for symptom relief in both MS and spinal cord injury. Although the current corpus of data does not definitively establish cannabis or particular cannabinoids as effective treatments for muscle spasticity, spasms, or pain in these conditions, it provides a crucial foundation for future research endeavors [149].

Indeed, given the inadequacies of existing treatment modalities for MS, a growing number of patients are exploring alternative therapeutic approaches, including cannabis extracts. There is increasing empirical validation for anecdotal claims of symptom alleviation through the use of cannabinoids, especially concerning muscle stiffness, spasms, neuropathic pain, sleep disturbances, and bladder issues. However, investigations targeting symptoms such as tremors and nystagmus have yet to yield favorable outcomes. In terms of safety profiles, cannabinoids generally appear to be well tolerated, with no major safety concerns reported during the testing phases. Moreover, improved tolerability has been observed with extended and gradual dosing regimens [150]. Recent advances in research methodologies and trial designs are actively being deployed to overcome existing limitations. Furthermore, burgeoning evidence suggests that cannabinoids may exert a modulatory influence on the core physiological processes pertinent to MS, including anti-inflammatory mechanisms, remyelination support, and neuroprotective functions. Consequently, ongoing clinical trials are investigating whether cannabinoids could not only provide symptomatic relief but also mitigate the progression of disability in MS patients, aligning with emerging insights in this research domain [150].

Emerging evidence suggests that the antispasticity effect observed in CREAE mice following treatment with AM374, an irreversible fatty acid amide hydrolase inhibitor, may be mediated through CB1/CB2 receptors. Studies have shown that the compound’s antispasticity impact is somewhat diminished with combined pre-treatment using SR141716A and SR144528, similar to findings involving R-(+)-WIN55212. However, the individual administration of SR141716A or SR144528 has been reported to exacerbate spasticity in CREAE mice. Despite these limitations, AM374 still holds promise for alleviating spasticity by stimulating endogenous CBRs. Additionally, it is noteworthy that multiple sclerosis patients using an oromucosal spray or oral DRO cannabis extracts can self-adjust dosages to minimize side effects without compromising therapeutic benefits. Moreover, there is evidence that the frequency of adverse events diminishes over the course of treatment without leading to a rapid adaptation to the medicinal effects of cannabinoids [151,152].

The utility of cannabinoids as immunosuppressive agents in chronic inflammatory conditions is further substantiated by research employing Theiler’s murine encephalomyelitis virus (TMEV), an animal model that mimics human multiple sclerosis. The administration of synthetic cannabinoids like WIN 55,212-2, ACEA, and JWH-015 during the established phase of TMEV infection yielded marked and sustained improvement in neurological deficits. Specifically, treatment with WIN 55,212-2, a non-selective CB1/CB2 agonist, led to a significant enhancement in RotaRod performance in TMEV-infected mice both immediately after treatment and 25 days post treatment. Similar outcomes were noted with ACEA, a selective CB1 agonist. Although the CB2 selective agonist JWH-015 did not achieve full restoration of motor function, significant improvement was still observed. Importantly, these functional gains persisted for at least 25 days after the cessation of treatment, underscoring the potential long-term benefits of cannabinoid therapy [153].

### 4.5. Mechanisms of Cannabinoid Action in Neurodegeneration

In the realm of neurodegenerative diseases such as Alzheimer’s, Parkinson’s, and Huntington’s (AD-PD-HD), neuroinflammation has been identified as a pivotal element contributing to neuronal degeneration. Various studies have demonstrated the efficacy of cannabinoids in ameliorating this inflammatory burden. For example, JWH015, a selective CB2 receptor agonist, has been shown to counteract the upregulation of CD40 in interferon-gamma-treated mouse microglial cells via interfering with the JAK/STAT pathway. This action further inhibits the production of proinflammatory cytokines while simultaneously promoting Ab phagocytosis. Additionally, compounds like CBD and the synthetic cannabinoids WIN 55212-2 and JWH-133 have been implicated in attenuating ATP-induced increases in intracellular Ca^2+^, a critical factor in microglial activation and the onset of inflammatory responses [111,134].

Acute neurodegeneration, such as the neurodegeneration resulting from cerebral ischemia due to stroke, trauma, or cardiac arrest, calls for immediate intervention. The existing research corroborates the neuroprotective properties of cannabinoids (CBs) in such scenarios, particularly in ameliorating secondary damage following the initial injury. Intriguingly, endogenous cannabinoids exhibit increased levels of production following brain trauma, suggesting a potential role in mitigating secondary injuries. For instance, levels of anandamide (AEA), an endocannabinoid, are elevated following controlled blood flow disruptions, a change that is attributed to the decreased expression and activity of the FAAH enzyme. Moreover, studies involving models of middle cerebral artery occlusion (MCAO) as well as clinical studies on stroke patients have corroborated the trend of elevated anandamide levels. Interestingly, levels of 2-arachidonoylglycerol (2-AG), another endocannabinoid, are noted to increase following physical traumas like concussive head injuries or seizures. However, its levels were found to decrease in mice subjected to MCAO-induced ischemia [83,154,155,156].

In the recent literature, cannabinoids have emerged as potent medicinal agents, particularly in the realms of appetite stimulation and antiemetic treatment for conditions such as cancer and AIDS. THC and CBD, the principal active compounds within the cannabinoid class, interact with G-protein-coupled receptors—CB1 receptors predominantly located in the CNS and CB2 receptors mainly found in immune cells. The body’s endogenous cannabinoids (ECBs), notably anandamide, virodhamine, and 2-arachidonoylglycerol (2-AG), bind to these receptors, facilitating various physiological responses such as cognitive function and pain perception. Elevated ECB levels have been observed in several neurodegenerative diseases, suggesting their potentially neuroprotective roles. Intriguingly, cannabinoids can function as agonists, antagonists, or inverse agonists when binding to CBRs, thereby modulating their neuroprotective effects. For instance, CBD has been shown to mitigate the toxicity caused by beta-amyloid peptides, resulting in reduced levels of reactive oxygen species, lipid peroxidation, and pro-apoptotic proteins. Additionally, CBD has demonstrated an ability to downregulate the production of proinflammatory cytokines and specific secretase enzymes. These attributes are promising for the use of cannabinoids in the treatment of neurodegenerative conditions like Parkinson’s and Alzheimer’s diseases [157] (Figure 2).

Over the past 15 years, scholarly attention has concentrated on elucidating the neuroprotective potential of agents that target the ECS, encompassing cannabinoid agonists, endocannabinoid degradation inhibitors, and allosteric modulators. These compounds have demonstrated a capacity to neutralize a variety of neurotoxic elements, including excitotoxicity, oxidative stress, and inflammation, thereby supporting neuronal health and longevity. Given the complexity of neurodegenerative disorders—which are often characterized by the simultaneous occurrence of multiple deleterious stimuli—an efficacious neuroprotective strategy necessitates a multifaceted approach to counteract these cytotoxic agents. In this context, cannabinoids distinguish themselves via their versatile neuroprotective properties. Unlike other compound classes under investigation for neuroprotective potential—such as antioxidants, N-methyl-D-aspartate (NMDA) receptor antagonists, and calcium channel blockers—cannabinoids offer a more comprehensive range of protective attributes. As a result, they have emerged as viable candidates for therapeutic interventions in conditions like stroke and traumatic brain injuries (TBI). It is imperative to note that most existing studies were conducted using animal models and often involve the administration of cannabinoids prior to the introduction of potentially cytotoxic factors. The relevance of such administration sequences to human pathology requires cautious interpretation. Among the various cannabinoids showing promise in preclinical models are Dexanabinol (HU-211), a synthetic compound with structural similarities to traditional cannabinoids but lacking cannabinoid receptor affinity; nonselective synthetic agonists like HU-210, WIN 55,212-2, TAK-937, and BAY 38-7271; and phytocannabinoids, including THC and CBD. The endogenous counterparts, such as 2-arachidonoylglycerol (2-AG) and anandamide, also hold significant therapeutic potential. These compounds have frequently been observed to confer a range of neuroprotective outcomes, including improved neurological function, diminished infarct sizes, and reduced edema and inflammation, alongside the modulation of immunomodulatory responses [158].

Cannabinoids demonstrate impressive antioxidative abilities by engaging the CB1 and CB2 CBRs to counteract free radical damage and modulate the production of ROS, as well modulating antioxidative defense mechanisms. The activation of CB1 receptors =initiates complex signaling pathways that play an essential role in supporting antioxidative responses and cellular survival, including the phosphoinositide 3-kinase (PI3K)/Akt, mitogen-activated protein kinase (MAPK), and Nrf2 pathways [159,160,161,162]. Furthermore, the activation of CB1 receptors governs key aspects of glutamatergic signaling such as activating the N-methyl-D-aspartate (NMDA)-receptor-activated regulation of calcium influx and the orchestration of Ca^2+^-dependent signaling cascades. CB2 receptors’ neuroprotective abilities derive from their capacity to reduce microglial activation and the release of pro-oxidative and proinflammatory agents. CB2 activation therefore plays a crucial role in mitigating neuroinflammation’s potentially damaging effects while also creating an environment that is conducive to cell integrity and sustained wellbeing. By harmonizing all of these intricate mechanisms, cannabinoids exert a multifaceted neuroprotective influence that promotes balance within an ecosystem conducive to sustained well-being [163] (Figure 3).

## 5. Cannabinoids and Cancer

### 5.1. Antitumor Effects of Cannabinoids

The investigation into cannabinoids as potential anticancer agents has expanded in recent years, although it remains relatively nascent. Limited to a small number of human studies, including one phase I/II clinical trial and three experimental studies, the body of evidence does reveal some promise. One of these studies distinguished itself through a rigorous methodological approach, aligning closely with the evaluation criteria outlined by the Cochrane Collaboration Manual. This meticulousness facilitated a more reliable interpretation of its experimental methodologies and outcomes, accentuating the need for further high-quality research to substantiate the antitumor effects of cannabinoids. Beyond merely serving as palliative agents in cancer treatment, cannabinoids hold potential as primary or adjunctive antineoplastic agents. However, the need for an expansive array of well-designed clinical trials remains critical for validating the antitumor efficacy of cannabinoids in oncological settings [164].

Recent advancements have particularly spotlighted the antiproliferative attributes of CBD, a nonpsychoactive cannabinoid. In a focused in vitro study examining the effects of CBD on U87 and U373 human glioma cell lines, a significant reduction in the mitochondrial oxidative metabolism was observed, along with a decrease in cell viability. The antiproliferative impact was noted within 24 h of exposure to CBD and was partially attenuated by specific agents like SR144528 and α-tocopherol. Intriguingly, other cannabinoid antagonists failed to reverse CBD’s effects. For the first time, the study linked CBD’s antiproliferative activity with the induction of apoptosis, which was confirmed via a cytofluorimetric analysis and single-strand DNA staining. Furthermore, in vivo studies on nude mice implanted with U87 human glioma cells demonstrated significant tumor reductions following the subcutaneous administration of CBD, reinforcing its potential role as an antineoplastic agent. These findings contribute substantially to our understanding of CBD’s antitumor properties, both in vitro and in vivo, advocating for its further exploration as a potential antineoplastic agent [165].

The therapeutic potential of Cannabis sativa, particularly its bioactive components like cannabinoids and terpenes, has garnered substantial attention in contemporary research. In a study involving female C57BL/6 mice treated with azoxymethane (AOM) and dextran sulfate sodium (DSS), THC exhibited both anti-inflammatory and antitumoral properties [166]. THC administration led to marked reductions in the severity of inflammation and tumor formation, as evidenced via the hematoxylin and eosin staining of the colonic tissue. Additionally, THC was found to mitigate the production of interleukin-22, a cytokine implicated in inflammation-driven colon cancer, by intraepithelial cells. Both cannabinoids and terpenes such as β-caryophyllene, limonene, and myrcene have demonstrated promise in inducing apoptosis, inhibiting cell proliferation, and suppressing angiogenesis in colorectal cancer (CRC). Of significance is the synergistic interaction between cannabinoids and terpenes, which may amplify therapeutic efficacy in treating CRC [167].

In a separate investigation focused on elucidating the antitumoral mechanisms of cannabinoid compounds, particularly those that are high in CBD, three extracts of Cannabis sativa were evaluated. The study centered on their effects on cell mortality, cytochrome C oxidase activity, and lipid composition in SH-SY5Y neuroblastoma cells. The results indicated that these extracts induce cell mortality by inhibiting the activity of cytochrome C oxidase. Importantly, this cytotoxicity was comparable to the cytotoxicity induced by known cannabinoid agonists like WIN55,212-2. While this effect could be partially attenuated by the selective CB1 receptor antagonist AM281 and antioxidants like α-tocopherol, it underscores the critical role of oxidative stress in mediating the antitumoral properties of cannabinoids. Furthermore, the extracts with high CBD contents revealed diverse antitumoral effects against human neuroblastoma cells which appeared to operate via multiple mechanisms, not only by affecting cannabinoid receptor activity but also by disrupting mitochondrial electron transport and increasing oxidative stress. Interestingly, the study suggested that whole-plant extracts may offer superior antitumoral effects compared to isolated cannabinoids. However, the study did not account for the potential mitigating impact of antioxidants, such as α-tocopherol. This omission is noteworthy since α-tocopherol, a well-known antioxidant commonly used to alleviate adverse reactions in chemotherapy, could potentially diminish the antitumoral efficacy of cannabinoid-based treatments [168].

### 5.2. Cannabinoids in Cancer Therapy

The expression levels of cannabinoid receptors (CB-Rs), particularly CB1-R and CB2-R, in breast cancer tissues have been illuminated through microarray technology analysis. The findings indicate that while CB1-R immunoreactivity was observed in 28% of carcinoma samples, a staggering 72% displayed CB2-R immunoreactivity. This is in stark contrast to non-transformed mammary tissues, which showed negligible immunoreactivity for both CB1-R and CB2-R. The association between elevated CB2-R expression and increased tumor aggressiveness is noteworthy. For instance, tumors devoid of estrogen and/or progesterone receptors, which generally have a poorer prognosis, frequently exhibit elevated levels of CB2-R. This trend is also seen in particularly challenging triple-negative tumors, which are characterized by their lack of both steroid hormone receptors and HER2/neu receptors. These tumors often display high CB2-R levels which correlate with poor differentiation, an increased likelihood of early local recurrence, and distant metastasis. The therapeutic landscape for breast cancer could potentially be revolutionized by targeting CB-Rs, particularly CB2-R and CB1-R. This avenue may offer effective treatment options for patients who experience recurrence post anti-HER2-targeted therapies. Beyond CB1-R and CB2-R, other CB-Rs like GPR55 also merit attention. The elevated expression of GPR55 has been observed in metastatic MDA-MB-231 cells, and its proliferative effects are thought to be linked to extracellular signal-regulated kinase (ERK) activation and the subsequent expression of the c-FOS proto-oncogene. Furthermore, cannabinoids (CBs) present potential therapeutic agents for challenging HER2-expressing breast tumors. Combining CBs with targeted therapies like lapatinib, a tyrosine kinase inhibitor, may potentiate antitumoral effects and enhance synergy with conventional chemotherapy agents such as cisplatin. Empirical studies have corroborated the synergistic effect between CBs and other oncologic agents including cisplatin [169,170,171].

From a translational standpoint, the synergistic potential of cannabinoids with existing chemotherapy treatments should not be overlooked. Preclinical studies have demonstrated that CBD and THC in particular can enhance the effectiveness of conventional chemotherapies. Although the scientific literature has yet to present data on the possible synergies between FAAH or MAGL inhibitors and classical chemotherapy or immunotherapies, cannabinoids have already been successfully employed in a clinical setting to mitigate the side effects associated with chemotherapy, such as nausea, vomiting, and pain. Recent work has also indicated the utility of MAGL inhibitors like MJN110 in reversing chemotherapy-induced neuropathy. Consequently, future research endeavors should prioritize combination studies with traditional chemotherapy agents to evaluate potential synergistic effects on tumor growth inhibition and metastasis reduction while simultaneously assessing the ability to alleviate chemotherapy-induced side effects [172,173].

In parallel, CBD has garnered an increasing amount of research interest for its analgesic properties in neurologically mediated conditions. One notable pharmacological formulation, Nabiximols (Sativex), which is a composite of CBD and THC, has gained regulatory approval in specific jurisdictions for mitigating spasticity associated with multiple sclerosis and as an adjunct in cancer-related pain management. CBD’s interaction profile is broad, encompassing not just the canonical CB1R and CB2R but also other receptors like TRPVs, 5-HT1A, GPR55, and PPARg. In the realm of oncology, CBD has exhibited anticancer properties through various mechanisms, including the induction of apoptosis and the inhibition of cell migration and metastasis across diverse cancer types [174].

Adding to the complexity of the cannabinoid landscape are compounds like cannabigerol (CBG), O-1602, and URB-602, which have shown promising anti-neoplastic effects in experimental models, notably in decreasing tumor volume and averting the formation of aberrant crypt foci (ACF) [175]. 

The ECS has emerged as a focal point of medical research owing to its regulatory role in an array of physiological and pathological processes, encompassing pain modulation and memory formation. Deviations in the activity of the ECS have been identified across a gamut of medical conditions, ranging from oncological to neurodegenerative disorders such as Parkinson’s disease, Huntington’s chorea, and multiple sclerosis (MS). Consequently, pharmacological interventions aiming to modulate the activity of the ECS have gained considerable momentum, often employing plant-derived or synthetic cannabinoids as active agents. Such pharmacological strategies have yielded tangible benefits in clinical contexts such as AIDS-related cachexia and MS-associated spasticity, among other palliative care applications. Prominent examples of these pharmaceutical agents include Sativex, a standard plant extract formulation of nabiximols, and synthetic compounds like Nabilone (Cesamet) and DRO (Marinol). While preliminary evidence suggests a potential utility of oral cannabinoids in ameliorating chemotherapy-induced nausea and vomiting (CINV), further empirical investigations are requisite to substantiate and consolidate this therapeutic application [176].

### 5.3. Potential Mechanisms of Cannabinoid-Mediated Anticancer Effects

Cannabinoids’ neuroprotective and antioxidant effects are produced via several complex mechanisms, the primary one of which is their effect on mitochondrial function. CB receptors typically reside on cell membranes. However, 30% of neuronal mitochondria contain CB1 receptors on their outer membranes, evidence that cannabinoids play an integral role in energy balance through the modulation of the mitochondrial electron transport chain (mETC), thus impacting learning processes as well as other physiological processes. The activation of the mitochondrial CB1 receptor pathway involves multiple components, including the Gai protein, soluble-adenylyl cyclase (sAC), and protein kinase A (PKA) [177,178]. Studies have also demonstrated that cannabinoids influence OXPHOS via non-receptor mechanisms, as supported by previous research [179]. Another study provides further evidence of a correlation between the inhibition of cytochrome C oxidase activity in SH-SY5Y cell lines and the concentration of THC in Cannabis sativa extracts and their ability to modulate the metabolism as well as the cannabinoids’ involvement in mitochondria-related toxicity and oxidative stress [168]. Cannabinoids’ production of ROS has been shown to cause changes to cell membranes, including the peroxidation of lipids that affect normal and cancer cells alike [180]. 

Limonene, a cyclic monoterpene found in citrus fruit peel oils, has been demonstrated to exert notable anticancer properties both in vitro and in vivo across various types of cancer. It can reduce tumor growth while simultaneously inducing apoptosis through multiple pathways. Limonene displayed significant cytotoxicity against T24 human bladder cancer cells by inducing G2/M-phase cell cycle arrest, decreasing migration and invasion, increasing apoptosis rates, and upregulating Bax/caspase-3 expression levels while attenuating Bcl-2 [181]. Limonene produced changes in gene regulation related to apoptosis, signal transduction, inflammation, and DNA repair within HepG2 cells. D-limonene demonstrated similar results in colon cancer cells, where it inhibited cell viability by inducing apoptosis through intrinsic pathway activation and suppressing PI3K/Akt activity [182]. For gastric cancer cells, however, the activation of the mitochondria-mediated intrinsic pathway was evidenced. Notably, the combination of limonene and berberine yielded amazing anticancer effects that surpassed their individual potencies [183]. Neuroblastoma cells were observed to exhibit autophagy through lipidated Light chain 3 (LC3) independent of the generation of ROS or ERK activation and in conjunction with decreased levels of p62 protein [184]. Lung cancer cell lines also displayed autophagy; D-limonene showed promising results at curtailing tumor growth in murine models [185]. D-limonene caused cell apoptosis through two distinct mechanisms in murine T-cell lymphoma cells: at lower concentrations, it caused the production of H_2_O_2_ and activated the ERK pathway, while at higher concentrations, it inhibited protein farnesylation and the production of O_2_ [186]. Niosomes containing 20uM D-limonene showed significant cytotoxicity against HepG2 cell lines as well as other cell lines; when combined with docetaxel, the effect was further amplified through an escalation in the production of ROS and an increase in the expression of apoptotic proteins, suggesting the involvement of the mitochondrial apoptosis pathway [187].

In an exploration of the mechanistic underpinnings of CBD and CBG treatments, one study focused on their impact on the expression of genes that are pertinent to cannabinoid activity and the pathobiology of mesothelioma. Notably absent from this analysis was CNR2 as its expression was not observed across any mesothelioma cell lines. The treatments with both CBD and CBG led to the substantial upregulation of key genes associated with cannabinoid activity and the pathology of mesothelioma across the three mesothelioma cell lines examined [188]. Specifically, noteworthy upregulations were observed for cannabinoid CB1 receptor (CNR1), G-protein-coupled receptor 55 (GPR55), and 5-HT1a receptor (5HTR1A) when compared to vehicle-treated controls, with an approximately 50-fold increase in the expression of GPR55. Interestingly, the CBD treatment had a variable impact on the mRNA expression of transient receptor potential vanilloid type 1 (TRPV1). While it influenced the expression of TRPV1 across most mesothelioma cell lines, an exception was noted in the case of H2452 cells [188]. The expression of TRPV2 or peroxisome proliferator-activated receptor gamma (PPARG) demonstrated cell-line-dependent variability. Moreover, CBD treatment led to a reduction in the endogenous CXCR4 agonist C-X-C motif chemokine 12 (CXCL12), while CBG’s effects on CXCL12 expression varied across different cell lines. To delve deeper into the mechanistic landscape, gene pathway analyses were conducted. Both CBD and CBG were observed to similarly influence cell cycle regulation pathways. Intriguingly, the Gaq/PLC signaling pathways may have been disrupted through the upregulation of GPR55 receptors by cannabinoids, affecting calcium homeostasis [189]. Furthermore, CBG appeared to stimulate the nuclear factor of activated T cells (NFAT) signaling pathways, a group of transcription factors that could also be activated via GPR55 receptors. Among the salient findings was the consistent activation of nuclear factor kappa-light-chain-enhancer of activated B cells (NF-kB) by both CBD and CBG. NF-kB is implicated in multiple inflammatory pathways, including the release of various cytokines and chemokines (such as CXCL12-CXCR4), along with cell cycle regulators, anti-apoptotic factors, and adhesion molecules. Overall, these preliminary results underscore the impact of CBD and CBG on human mesothelioma cell lines, indicating a clear avenue for further investigation [190].

In a subsequent investigation, a research team explored the antitumoral potential of WIN 55,212-2 against pediatric osteosarcoma. The study delineated that WIN 55,212-2 induced cell cycle arrest and prompted the upregulation of crucial markers of endoplasmic reticulum stress such as GRP78, CHOP, and TRB3, followed by autophagy [191,192,193]. These findings align with previously reported mechanisms in adult cancers. For example, Fisher et al. conducted a study on the impact of both THC and CBD on pediatric neuroblastoma cells, revealing significant reductions in cell viability. Additionally, CBD was observed to inhibit xenograft growth in vivo. Although the precise mechanisms underlying CBD’s antitumoral effects remain to be elucidated, the compound induced apoptosis in neuroblastoma cells both in vitro and in vivo, effectively causing cell death directly and indirectly [194].

Collectively, these preclinical findings demonstrate the prospective anticancer efficacy of cannabinoids against a range of pediatric cancers, albeit via multiple mechanisms. It is important to recognize that pediatric cancers are heterogeneous, originating from various cell types and tissues and often driven by specific mutations. One limitation of these studies is their reliance on long-term cultured cell lines that may not fully represent the complexity of human cancers. Furthermore, a dearth of studies have corroborated their findings through animal models or orthotopically xenografted models, consequently failing to replicate the authentic tissue contexts of these malignancies [195]. Moreover, no clinical trials have assessed the potential antitumoral effects of cannabinoids in the treatment of pediatric cancer. Anecdotal evidence exists, however, suggesting potential benefits. For instance, Foroughi et al. reported two cases of female patients experiencing spontaneous regressions of low-grade glioma coinciding with cannabis inhalation. While a retrospective study linked the expression of CB1R to tumor regression, suggesting a plausible mechanism, Foroughi et al. did not investigate CB1R expression in their reported cases [196,197]. 

A growing body of literature supports the notion that the therapeutic benefits of cannabis are not merely attributable to individual constituents but are the result of synergistic interactions among various compounds within the plant. For instance, evidence has emerged that a holistic botanical preparation of cannabis exhibits greater potency in both in vitro and in vivo models for breast cancer treatment compared to isolated 9-THC [198]. Furthermore, in vivo experiments yielded compelling results, showing additive effects when cannabis terpenes such as α-humulene and β-pinene were combined with WIN55,212-2 in mouse models [199]. These findings suggest that the inclusion of terpenes enhances the activity of isolated cannabinoids, likely through a synergistic mechanism. Parallel lines of inquiry have delved into the molecular mechanisms underlying potential pharmacological interventions against prostate cancer. Various prostate cancer cell lines, including PC3, DU145, and LNCaP, have demonstrated reduced migration, implicating cannabinoids as promising agents in combating cancer cell motility. Specifically, the CB1 agonist WIN-55,212 was observed to decrease the activity of RhoA GTPase, a critical regulator of cell migration [200]. This led to a subsequent disruption in actin/myosin microfilament formation and a reduction in cell migration. Reinforcing RhoA protein activity resulted in an increase in microfilament formation and cell spreading, whereas the exogenous CB1 agonist anandamide mimicked the reduction by disrupting actin/myosin microfilaments. In this context, Roberto et al. reported significant, dose-dependent decreases in the migration and invasion capabilities of PC3 and DU145 cells when treated with the synthetic cannabinoid WIN-55,212 [201].

## 6. Clinical Applications and Challenges

### 6.1. Cannabinoids as Therapeutic Agents

Emerging combined cancer therapies have generated widespread interest in the use of cannabis botanicals for treating glioblastoma (GB). Their polypharmaceutical nature offers distinct advantages over current therapies. They may complement standard-of-care treatments more effectively by fully harnessing their anticancer properties. Moreover, they have off-target effects that are less toxic than those of traditional chemotherapeutics. The use of cannabinoids has already proven successful as palliative care in many GB patients. Cannabinoids as anticancer agents can be seen in numerous academic publications, demonstrating tumor-specific cytotoxic and cytostatic effects in experimental models as well as clinical studies, including those on GB patients [202]. Furthermore, GB stands out in that its aggressive infiltration within the brain parenchyma limits metastatic spread outside it; this paradoxical behavior underscores both its complexity and resistance to treatment. Cancer stem cells (CSCs) play an essential role in the resistance of GB to therapy, with active DNA repair mechanisms and efficient xenobiotic export systems being key determinants [203]. Therapy-resistant CSCs may lie dormant in protective niches before becoming aggressive cells that trigger tumor regrowth elsewhere in the brain. The findings of Lah et al. show that all three cannabinoids induced a significant apoptotic rate of approximately 30% among GB cancer stem cells at their respective IC50 concentrations, suggesting the significant inhibition of cell viability mechanisms as cytotoxins, with the significant inhibition of cell viability mechanisms being a key mechanism of their cytotoxicity and the inhibition of viability mechanisms. Notable among CBG’s signaling effects is the activation of caspase-3/7, which is further amplified with CBD and temozolomide (TMZ) [204]. 

Experiments using a spinal nerve ligation neuropathic pain model revealed that administering CB1 receptor-selective agonist ACEA led to decreased mechanically evoked responses in spinal neurons. This effect could only be prevented with CB1 antagonists such as AT1077. When applied systemically and locally in rats suffering chemotherapeutic-agent-induced neuropathic pain, both the systemic and local administration of ACEA demonstrated attenuated mechanical allodynia without inducing psychoactive side effects at the administered doses [205].

Another compound, the CB1/CB2 dual agonist CRA13, demonstrated powerful anti-hyperalgesic properties in an animal model of neuropathic pain. Both the oral administration and local injection of CRA13 were effective at reversing mechanical hyperalgesia caused by established mechanical hyperalgesia. Importantly, its anti-hyperalgesic action occurred via peripheral CB1 receptors, as evidenced by its responsiveness to CB1 antagonist [206]. Preclinical studies showed AZD1940, an orally active mixed CB1/CB2 receptor agonist, to have an analgesic effect in both inflammatory and neuropathic pain models without leading to the development of tolerance or a high level of brain uptake at effective antinociceptive doses in rats or primates [207]. Its analgesic action was CB1-receptor-dependent, acting peripherally without leading to the formation of tolerance; the brain uptake at effective antinociceptive doses was low. However, clinical studies of AZD1940 yielded mixed results. While preclinical trials demonstrated promising effects against capsaicin-induced pain and hyperalgesia in human trials, its analgesic properties failed to reduce post-operative dental extraction pain in healthy subjects, and mild-to-moderate gastrointestinal and CNS side effects were reported in clinical studies of this compound [207,208,209]. While certain compounds showed promising analgesic and peripheral-site-of-action effects in animal studies, their translation into human clinical trials yielded variable outcomes and side effects. Furthermore, harnessing cannabinoids as pain management solutions is both possible and difficult due to how intricately connected CBRs and pain pathways are.

Cannabis has long been used to treat various conditions, and one such ancient use was to manage epilepsy. Epilepsy, a chronic neurological condition affecting millions worldwide, is characterized by recurrent seizures which are often coupled with cognitive impairments and mood disturbances. Efforts at management typically revolve around modulating neuronal ion channels as well as GABA/glutamate receptors, yet approximately one-third of epileptic patients remain resistant to the current treatments available to them [210]. Cannabinoids may help alleviate epileptic seizures due to the presence of CB1 receptors in key brain areas involved in partial seizure initiation, such as the hippocampus and amygdala [211]. The studies conducted to date have highlighted the significant anticonvulsive properties of various cannabinoids, especially CBD, and more recently, CBDV/D9-THCV [212]. Unfortunately, however, its exact antiepileptic mechanisms remain unknown due to its relatively low affinity for CB1 and CB2 receptors. CBD may exert its effects through several pathways, including interactions with the equilibrative nucleoside transporter and GPR55, TPRV-1, and 5-HT1A receptors, as well as the a3 and a1 glycine receptors [213,214]. Another possible antiepileptic mechanism of CBD could involve its interactions with mitochondrial Na^2+^/Ca^2+^ exchanger [215]. 

CBD’s therapeutic potential goes well beyond epilepsy, with applications across a broad range of both nonpsychiatric and psychiatric disorders including anxiety, depression, bipolar-disorder psychosis, and sleep disturbances, and a significant amount of research into its pharmacological effects across various biological systems has been undertaken to understand its mechanisms of action as medicine. Animal models suggest that CBD may produce anxiolytic-like effects by activating post-synaptic 5-HT1A receptors located in key brain regions associated with defensive responses, such as the dorsal periaqueductal grey dorsal periaqueductal grey bed nucleus of the stria terminalis and the medial prefrontal cortex [216].

Due to opioids’ shortcomings and risks, cannabis’ potential as an effective pain remedy has garnered increasing consideration. Legalizing cannabis has proven its safety potential by leading to a decrease in opioid overdose deaths. In general, healthcare providers have shown an encouraging outlook towards the therapeutic uses of cannabis for patients, actively helping to facilitate access to medical cannabis. Patient perspectives support this sentiment, with many believing in cannabis’ effectiveness as an analgesic and considering it an alternative medication to opioids. Some patients perceive cannabis to be both safe and effective for treating multiple medical conditions, prompting their cannabis use. Sometimes patients combine cannabis use with prescription drugs; the impact on overall well-being remains uncertain in such instances [217]. 

Secondary metabolites derived from both cannabinoids and non-cannabinoids have vast therapeutic potential across a wide range of conditions, from cancers, diabetes, cardiovascular issues, neurodegenerative disorders, inflammatory diseases, and viral infections to neurodegeneration disorders and viral infections. Unlike THC—one of the best-studied cannabinoids—most of these phytochemicals lack psychotropic effects, enabling them to provide therapeutic benefits without creating the psychoactive responses associated with THC [218].

### 6.2. Clinical Trials and Evidence-Based Medicine

Until now, limited and disparate research findings have illuminated the effects of cannabinoids on mental health during the prepubertal stages. Studies have demonstrated that providing CBD during peri-pubertal periods may reduce the behavioral abnormalities seen in animal models of schizophrenia. Preclinical evidence also shows that exposure to both THC and stress during peri-adolescence could result in impaired fear extinction in adulthood for mice, though this was not evident among animals who only received either THC or stress alone [219]. Therefore, further clinical investigations must be conducted in order to ascertain whether concurrent exposure to cannabis and stress during teenage years might contribute to long-term anxiety disorders or pathological fear in adulthood. Studies have demonstrated that frequent cannabis users may suffer neurocognitive deficits, including reduced psychomotor speed and working memory, but these effects can be effectively and affordably improved through aerobic fitness activities—showing the potential of physical fitness to ameliorate the cognitive deficits associated with cannabis consumption in adolescents [220,221,222].

Multiple sclerosis (MS) spasticity therapy aims to increase functional capacity, facilitate rehabilitation, prevent contractures, and alleviate discomfort among individuals diagnosed with MS. Cannabinoids stand out as notable interventions within neurological disorders, particularly MS. Cannabinoids have proven to be particularly successful at managing MS-related spasticity in several recently conducted studies, demonstrating its benefits among complementary medicine approaches like pharmaceutical cannabinoids. Nabiximols has emerged from multiple rigorous randomized controlled clinical trials against placebos to gain approval as an effective medication for alleviating spasticity-related symptoms. Notably, one recent enriched-design methodology study demonstrated that adding Nabiximols provided more effective relief from MS spasticity than simply adjusting an anti-spasticity medication regimen. Another investigation known as SAVANT explored the use of oromucosal Nabiximols as adjunctive therapy against moderate-to-severe spasticity symptoms [223,224,225,226,227]. 

Studies exploring means of reducing opioid doses for managing chronic pain may not always produce reliable findings due to the incomplete reporting of dose adjustments and analgesic outcomes. Notable recent analyses have not yielded evidence that cannabis exerts any opioid-sparing properties. Preclinical evidence points toward cannabinoids’ effectiveness for treating inflammatory bowel diseases; preclinical findings demonstrate CBD’s protective impact against intestinal inflammation. Yet more rigorous clinical trials must still be conducted on a larger scale to ascertain whether cannabinoids or their derivatives offer any advantages when treating individuals afflicted with IBDs [228,229].

In a landmark interventional pilot clinical trial, the first of its kind to document anti-inflammatory effects following cannabinoid administration in humans, a focus was placed on individuals living with HIV (PWH) and undergoing antiretroviral therapy (ART). The study successfully completed treatments involving eight participants who were administered oral cannabinoids. The results displayed significant reductions in surrogate markers linked to gut mucosal damage, systemic inflammation, immune cell activation, fatigue, and cellular senescence. These initial outcomes advocate for extended investigations through larger clinical trials, aiming to determine the feasibility of using cannabinoid capsules to mitigate chronic inflammation in PWH on ART [230]. Within the scope of this pilot trial, safety and tolerability were the principal concerns. Additionally, the study probed the impact of oral cannabinoids on the integrity of the gut mucosal barrier. This was achieved by monitoring the dynamics of REG-3a and I-FABP throughout the treatment period. REG-3a is instrumental in modulating interactions between humans and gut microbiota, whereas I-FABP is released upon the death of enterocytes. Elevated plasma levels of these markers were previously noted among pregnant women on ART; however, these levels were observed to decline following a 12-week regimen of oral cannabinoid therapy. This corroborates the results of earlier studies elucidating the beneficial impacts of CBD and palmitoylethanolamide on gut mucosal permeability, which are attributed to CBD’s activation of CB1R in the ECS [231]. Although substantial anti-inflammatory benefits were evident in the trial, the precise mechanisms underlying these effects remain unclear. An additional layer of complexity is introduced by the dual nature of THC as both a partial agonist and antagonist for CB1R. Furthermore, THC can interact with other endocannabinoid receptors to exert anti-inflammatory effects [230,232]. 

### 6.3. Safety Considerations and Adverse Effects

Notable precautions regarding cannabis and THC revolve around their neuropsychiatric side effects, which often determine their maximum tolerated dose and can result in discontinuation. Clinical trials conducted using DRO have demonstrated its potential to exacerbate conditions like mania, depression, and schizophrenia. Recent meta-analyses conducted on cannabinoids demonstrated an almost threefold increased likelihood of experiencing adverse mental or nervous system effects compared to comparator groups, although individual symptoms such as anxiety or depression did not show statistically significant variations [233]. The FDA recommends pre-screening patients before initiating THC therapy as well as medical cannabis treatments; pre-screening should extend across both treatments. Furthermore, THC has the potential for the development of a dependency among individuals with histories of substance use disorders involving nicotine, alcohol, opioids, or illicit drugs [234]. While recreational users may seek certain effects specifically from THC use, clinical trial participants have reported adverse side effects like disorientation, dissociation, euphoria, and hallucinations, which can pose particular dangers to medically vulnerable populations such as older adults [235]. Recent prescribing information for newly approved products has highlighted the elevated risk of suicidal behavior and ideation associated with psychoactive medications, such as antihypertensives, antidepressants, and opioids [236]. As with CBD recommendations—though with more emphasis being given to THC-containing products—individuals experiencing depression or those who are taking medications which share similar risk should exercise extreme caution when considering medical cannabis products as alternatives. Antipsychotic and antidepressant users should prioritize selections with reduced potential for drug—drug interactions (DDIs) in keeping with the principle of prudent medication management.

Cannabinoids are widely perceived as harmless substances by the general population, and any long-term health implications are overlooked. Comparing cannabinoid users and non-users within the broader population highlights their potentially negative impact on cognitive function. There is an increasing amount of evidence linking acute cannabinoid use to deficits in neurocognitive decision making across areas like processing speed, sustained attention span, verbal fluency, and executive functioning. Over time, chronic cannabinoid consumption among teenagers and young adults has shown adverse impacts across various cognitive domains like learning memory, attention, executive function, and psychomotor speed [237,238]. 

Cannabinoids’ growing prevalence, both recreationally and clinically, has increased the possibility of the co-administration of cannabinoids with selective serotonin reuptake inhibitors (SSRIs), potentially leading to adverse outcomes. A comprehensive analysis was performed on adverse event reports submitted through the FAERS of the U.S. Food and Drug Administration’s Adverse Event Reporting System; the results demonstrated significant instances in which cannabis or its derivatives caused adverse events that demonstrated an interaction risk between this substance and other medications. Notably, adverse events reported have shown an upward trend over time due to the increased availability of marijuana-derived products, both prescription and over-the-counter (OTC) THC/CBD for medical and recreational use. Although direct clinical interactions remain an evolving area of research, one case report has hinted at an association between cannabis hyperemesis syndrome and the concurrent use of an SSRI medication [239]. By analyzing a vast dataset encompassing nearly 15 million patient reports from the FDA, specific cohorts were formed in order to analyze the frequencies of side effects. This included medications (sertraline, escitalopram, and citalopram), cannabinoids (THC, CBD, and other cannabinoids) and combinations which were metabolized via CYPC219 [240]. The frequencies of sertraline side effects served as baseline for comparison against the cannabinoids/combinations cohorts; 23 side effects each, with an occurrence rate exceeding 5% on sertraline or escitalopram labels, were selected for benchmarking purposes. Established pharmacovigilance metrics, relative risks, and safety signals were employed to identify potential associations between drug or combination side effects and strict statistical methodologies, such as Benjamini–Hochberg–Yekutieli tests, to ensure accurate statistical significance with a threshold set at 0.05 while also compensating for the false discovery rate (FDR) [241]. As is evident by an increase in adverse events associated with the co-administration of cannabinoids and SSRIs, careful consideration must be paid when researching potential interactions between them [240].

## 7. Future Perspectives and Conclusions

### 7.1. Promising Avenues for Future Research

THC interacts with the ECS in an interesting fashion by acting as both a partial agonist of CB1 and CB2 receptors and an agonist for GPR55 receptors, while CBD acts as an antagonist or negative allosteric modulator of these same receptors, leading to its ability to modulate THC’s psychotropic effects when co-administered. While CBD displays weak binding to CB1 and CB2 receptors at therapeutic doses, its influence on GPR55 curtails intracellular calcium release, thus potentially mitigating the neuronal hyperactivity associated with conditions like epilepsy. The administration of CBD has been linked to elevated serum levels of anandamide (AEA), potentially contributing to its therapeutic effects in schizophrenia patients [242]. Unfortunately, however, its exact mechanism for elevating AEA levels remains incompletely understood and requires further study. Some evidence suggests that both THC and CBD preferentially bind fatty-acid-binding proteins that are essential for the intracellular transport of anandamide and its subsequent degradation via FAAH within cells, contrary to rodent studies which demonstrated an inhibition of FAAH activity. Such results highlight the limitations of animal models as research tools in cannabinoid research [243]. Cannabinoids such as THC and CBD exhibit potency anti-inflammatory actions through COX-2 inhibition, leading to a reduced production of pro-inflammatory prostaglandins [244]. This may indirectly raise endocannabinoid levels, thereby contributing to their antiepileptic properties. CBD’s influence on CYP isoenzymes in the brain may further modulate the production of specific eicosanoids like EETs, EET-EAs, and HETE-EAs which may exert indirect influence upon receptors via the regulation of downstream eicosanoid pathways; its influence on CYP isoenzymes, in addition to its inhibition of their production, show its intricate interaction with various molecular pathways [245]. CBD’s ability to decrease 5-LOX activity and its related metabolites in human tumor cells raises the possibility that CBD could have anti-seizure properties [246]. This possibility is particularly intriguing given the possible link between the targeted inhibitors of Cys-LT synthesis and reduced seizure risk, although further validation will likely be required before this can be established [247].

Regarding anxiety modulation, due to their widespread presence in key regions of the brain associated with emotional responses such as the prefrontal cortex, amygdala, and hippocampus, CB1Rs have attracted increased scrutiny as potential sources for reducing anxiety. CB1R has been linked with controlling behavioral responses associated with altered emotional states [248,249]. Studies have demonstrated that cannabinoid agonists exhibit a two-pronged approach to controlling anxiety and stress, with lower doses showing attenuated effects, while higher ones could induce anxiety-inducing responses. CB1R-mediated responses involve diverse molecular mechanisms, including contradictory roles played by different areas in modulating anxiety, the activation of specific CB1R populations on GABAergic or glutamatergic neurons, and the potential engagement of non-CB1R-related pathways [250,251,252]. Environmental factors, including stress-induced changes in GABA responses to CB1R agonists, also play a part. CB1Rs are found not only on GABAergic and glutamatergic neurons but are also present in raphe nuclei to influence serotonergic nerve cell function. Studies involving mice lacking CB1R-deficient serotonergic neurons have revealed increased anxiety levels and diminished socialization skills. Neuroinflammatory processes have long been implicated in anxiety and depression. Both CB1R and CB2R possess the power to alleviate neuroinflammation, offering multiple avenues for anxiolytic effects. Notably, CB2R polymorphisms in Japanese individuals with depression have been linked to altered behavior. The antisense oligonucleotide targeting of CB2R mRNA expression in mice led to reduced anxiety-like behaviors. Studies involving CB2R-overexpressing mice and spontaneously anxious mice have indicated that CB2R can play an essential role in modulating anxiety-like behaviors through its interactions with GABA receptors. The deletion of CB2R from dopaminergic neurons located in the ventral tegmental area (VTA) of mice was demonstrated to significantly affect anxiety, depression, and psychomotor behavior. Given the adverse neurological reactions associated with CB1R antagonism by rimonabant, other therapeutic avenues involving CB2R modulation may offer promising solutions [253,254,255,256,257].

### 7.2. Implications for Cannabinoid-Based Therapies

Immunity plays an integral role in protecting against foreign agents and pathogens, so for over four decades, researchers have investigated how cannabinoids impact immune responses against pathogens. Notably, in 1977, a groundbreaking study by Bradley et al. demonstrated how THC combined with lipopolysaccharide (LPS) caused increased toxicity while amplifying the lethality of heat-killed bacteria [258]. Subsequent investigations by this same group explored the effects of THC and cannabis extracts on host resistance to Listeria monocytogenes and herpes simplex virus, ultimately showing decreased levels of pathogen resistance in subsequent investigations [259,260]. Subsequent studies revealed the roles of the ECS in initiating immune responses against pathogens, with specific CB2 genotypes correlating with susceptibility to certain viral illnesses. Conversely, in vitro studies demonstrated the microbicidal activity of cannabinoids against various bacteria and fungi, as well as some instances of the regulation of viral pathogenesis by cannabinoids [261]; moreover, the oral administration of cannabis increased survival in murine models of malaria with enhanced host immunity, while increased levels were detected within their lungs and intestines [262]. This was further evidenced by increased host immunity, which was observed with elevated endocannabinoid levels which increased survival when orally administered to infected animals, indicating its protective nature. Cannabinoid-based treatments for infectious diseases are determined by two key elements: their anti-inflammatory and pathogen-targeting capabilities [263,264]. 

Another intriguing area is vaccination and how Cannabis/CBD treatments impact vaccine-related immunity. Dotsey et al. explored this connection using a transient CB2 blockade on immune reactions of young and aged mice undergoing vaccination; intensified antigen-specific immune reactions were noted following immunization [265]. However, a prospective study of humoral/cellular immune responses during hepatitis B vaccination among habitual marijuana smokers did not reveal significant alterations to the development of systemic immunity [266].

Inflammatory bowel disease (IBD) represents a category of autoimmune gastrointestinal disorders that includes ulcerative colitis (UC) and Crohn’s disease (CD). Evidence from various studies has substantiated the anti-inflammatory potential of cannabinoids in mitigating colitis in murine models. Interestingly, these effects were found to be reversed when CBRs were either blocked or deficient. In the realm of preclinical research, phytocannabinoids have been employed in models of gastrointestinal inflammation, and their efficacy has further been examined in clinical trials involving IBD patients [267]. A study utilizing a DSS-induced murine model of colitis revealed a particular sensitivity to cannabinoid-based interventions. Intriguingly, cannabis extract treatments outperformed their pure-cannabinoid counterparts, possibly due to the synergistic interactions and distinct anti-inflammatory attributes conferred by other phytochemicals present in the whole plant [267]. Macrophages, which are aberrantly regulated in IBD, play a pivotal role in the pathogenesis of the disease. These cells are notably abundant in the inflamed mucosa of IBD patients and display an altered phenotype and function compared to normal conditions, such as the elevated expression of co-stimulatory molecules and the production of the inflammatory cytokines IL-12 and IL-23. The research indicated that cannabinoid-based treatments were effective in inhibiting the infiltration of macrophages into the colons of DSS-induced mice, with the severity of the disease directly correlating with the average number of infiltrating macrophages. Moreover, different cannabinoid treatments had varying effects on cytokine levels in murine models. CBD was observed to reduce interferon-beta and interleukin-6 levels, whereas THC primarily diminished interleukin-6 levels. CBDE exerted an impact on TNF-alpha and IL-6, while THCE significantly curtailed the levels of all three investigated cytokines. Importantly, all the cannabinoid treatments examined influenced IL-6 and TNF-alpha, key cytokines associated with IBD. CBDE emerged as particularly effective in downregulating TNF-alpha levels, which is noteworthy considering the frequent clinical usage of anti-TNF agents due to the cytokine’s central role in disease pathogenesis in both human and murine models [267,268].

In a comprehensive analysis aimed at evaluating the impact of cannabinoids on lymphocyte function, in vitro methodologies were employed. The study particularly focused on cannabis extracts enriched in CBD Botanical Drug Substance (CBD BDS) or THC Botanical Drug Substance (THC BDS), with concentrations ranging from 20% to 30% of each. The inclusion of these cannabis extracts alongside pure cannabinoids served a dual purpose. Firstly, it acknowledged the prevalence of cannabis-based medicines over isolated cannabinoids in patient care. Secondly, it allowed for an exploration of the putative advantages conferred by the entourage effect. Activated lymphocyte proliferation was the primary outcome measure under investigation. Anti-CD3 antibodies were employed to activate mouse splenocytes (C57BL/6 or BALB/c), which were subsequently exposed to various concentrations of pure cannabinoids or their botanical drug substance counterparts. An FACS analysis was then utilized to assess cell proliferation. Interestingly, it was revealed that the pure cannabinoids were more potent in inhibiting lymphocyte activation compared to cannabis extracts. Among the pure cannabinoids, CBD was found to be more effective at curtailing proliferation than THC irrespective of the form in which it was administered [269]. Similar outcomes were also observed in human peripheral blood mononuclear cells (PBMCs). Upon the activation of anti-CD3, an increase in the CD8 cell percentage was noted, an effect that could be efficaciously mitigated by CBD, CBD BDS, and, to a lesser extent, THC BDS treatments [269]. 

### 7.3. Concluding Remarks on the Potential of Cannabinoids

Cannabis is an impressively complex plant, boasting more than 100 cannabinoids in addition to various terpenes and flavonoids. Our understanding of its effects is further complicated by cannabinoids’ demonstrated activity across numerous receptors. This characteristic bestows cannabinoids—and by extension, cannabis itself—with the label of being promiscuous drugs; though often seen as disadvantageous, the promiscuity of a drug actually provides distinct advantages, the most important of which is the ability to engage various pathways within an illness with just a single therapeutic agent. Medical cannabis has seen a rapid expansion in recent years as more patients turn to using it as a solution for various ailments. With more patients turning to this botanical remedy for treatment purposes, a growing demand exists among the scientific and medical communities to investigate how cannabis orchestrates its effects within the body; this goes beyond simply understanding potential merits and risks. Optimal routes of administration vary according to each condition and must also be investigated carefully. As the use of medical cannabis continues to expand, research initiatives become ever more necessary. These endeavors must not only dissect how the components of cannabis interact within the body but also establish safe routes of application for various medical scenarios. Furthermore, an understanding of all of its complexities must be obtained before effective use can occur within clinical environments [270].

Cannabinoids and eCBs have quickly become one of the most exciting areas of biomedical and chemical research, witnessing over 1000 publications annually with an expected upward trajectory. The investigation into cannabinoid delivery systems has also witnessed tremendous activity, with companies filing patents relating to localized or transdermal administration. Innovative formulation strategies provide an effective avenue for producing swift systemic effects with sustained long-term effects, as evidenced by the potential synergy between intranasal cannabinoid sprays and patches for fast absorption and an immediate systemic impact. Furthermore, compelling findings include using terpenes from cannabis sources (CBD and THC) as penetration enhancers to increase the efficacy of therapeutic constituents, further underscoring quality control’s important role in shaping the composition, dosage, and safety profiles of cannabis-derived components. Amid today’s evolving therapeutic paradigms are possibilities for innovative therapeutic paradigms combining established cannabinoids in new applications with engineered cannabinoid derivatives. Nanotechnology presents particularly encouraging avenues, with SEDDS representing one promising route towards realizing the clinical use of Cannabis, involving both oral and pulmonary routes of administration [271]. The integration of carbon nanotubes is still in its infancy but holds promise as an efficient delivery system; however, thorough scrutiny is required to optimize cost-effectiveness and the long-term safety of nano-delivery systems before adopting them into mainstream applications. As the surge in interest in cannabinoids coincides with advances in pharmacology, pharmaceuticals, and technology, an enabling environment is set up for the creation of innovative therapeutic strategies that leverage established cannabinoids and their synthetic derivatives. Science meets innovation while quality and safety issues come together to shape cannabinoid-based interventions for therapy in the near future.

## Figures and Tables

**Figure 1 biomolecules-13-01388-f001:**
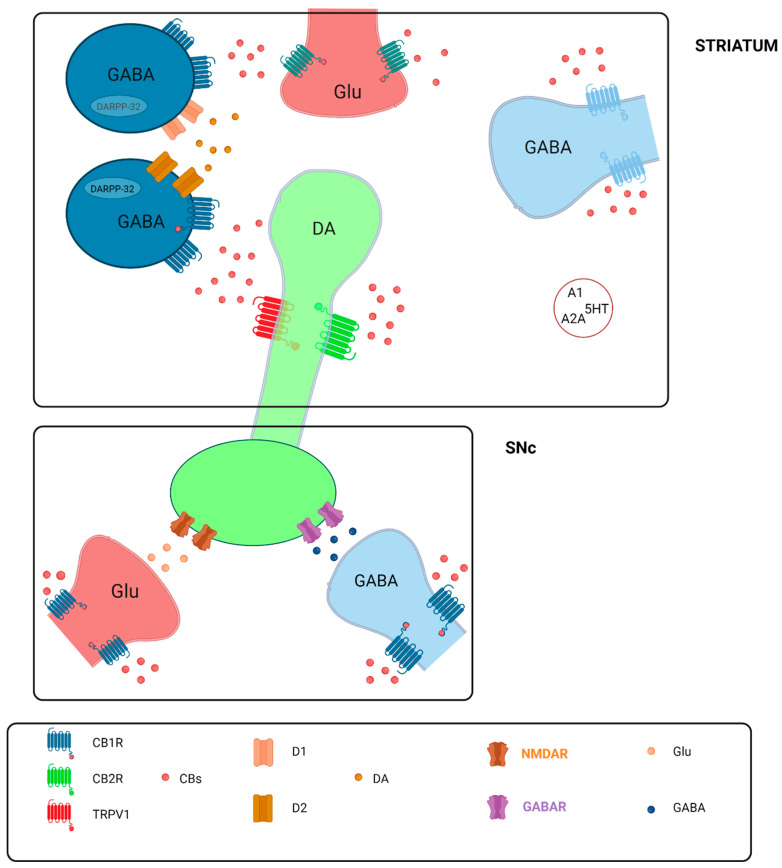
The distribution of the CB1R and CB2R in the striatum of the rat.

**Figure 2 biomolecules-13-01388-f002:**
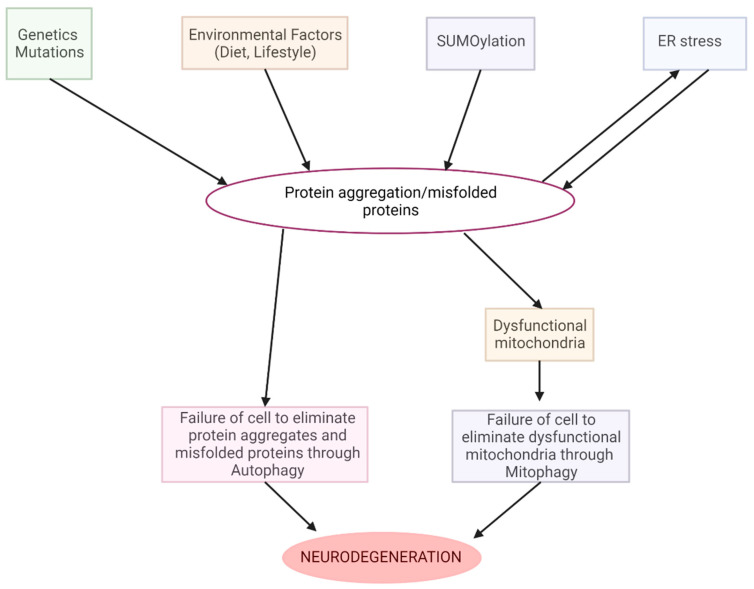
The mechanism of the protein aggression and degeneration that lead to neurodegenerative phenomena.

**Figure 3 biomolecules-13-01388-f003:**
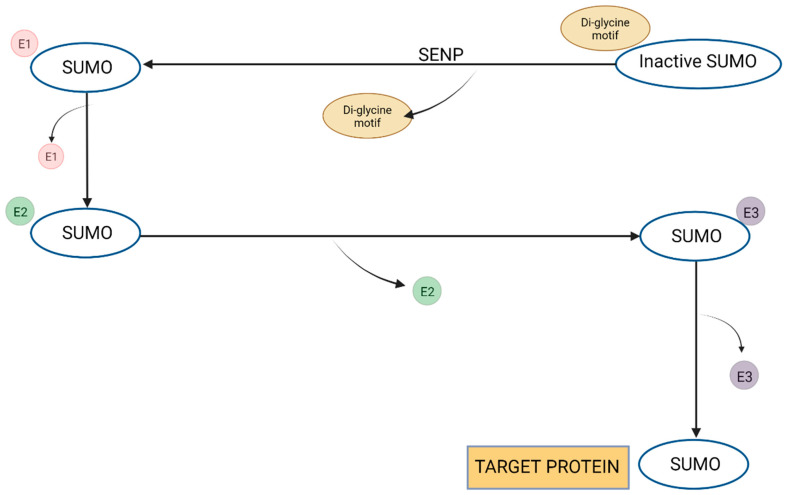
The enzymatic steps leading from inactive SUMO to activated SUMO.

**Table 1 biomolecules-13-01388-t001:** A summary of molecular changes occurring in R6/1 and WT mice at 20 wk of age following chronic cannabinoid drug treatment compared to vehicle treatment.

	Brain Region	Drug	R6/1	WT
Aggregate number	Striatum	HU210	Increased	N/A
CB1 ligand binding	Striatum	URB597	Increased	None
	Hippocampus	THC	None	Decreased
CB1 mRNA	Striatum	HU210 URB597	None	Decreased
GABA_A_ ligand binding	Globus pallidus	URB597	None	Increased
5HT2A ligand binding	Striatum	HU210 URB597	None	Decreased
	Hippocampus	URB597	None	Decreased
	Motor cortex	URB597	None	Decreased

**Table 2 biomolecules-13-01388-t002:** Cannabinoids and neurodegenerative diseases.

	CB_1_ Receptor	CB_2_ Receptor	Endocannabinoid Levels	Endocannabinoid Synthesis	Endocannabinoid Degradation
Alzheimer’s disease	CB_1_ receptor expression initially increased, followed by a decline during disease progression [140]. CB_1_ receptor was functionally impaired [141].	CB_2_ receptor increased in the entorhinal cortex and parahippocamus [142].	Decreased AEA levels in the midfrontal and temporal cortex [143].	DGLalfa and DGLbeta levels were increased in AD patients (Braak stage IV) [144].	Increased FAAH levels [145]. Increased MGL levels in AD patients (Braak stage IV) [144].
Parkinson’s disease	CB_1_ receptor expression decreased in the substantia nigra. CB_1_ receptor expression increased in dopaminergic projecting areas [146]		AEA levels increased in cerebrospinal fluid. A sevenfold increase in 2AG levels in the globus pallidus [147].	-	Decreased levels of anandamide membrane transporter and FAAH [145].
Huntington’s disease	CB_1_ receptor expression decreased in the caudate nucleus, putamen, and globus pallidus [148].	CB_2_ receptor expression increased in striatal microglia [137].	AEA and 2AG levels decreased in the striatum. AEA levels increased and 2AG levels decreased in the cortex [145].	NAPE-PLD and DGL levels decreased in the striatum [145].	FAAH levels increased and MGL levels decreased in the cortex [145].

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
