# Peer review of "Cannabinoids in Medicine: A Multifaceted Exploration of Types, Therapeutic Applications, and Emerging Opportunities in Neurodegenerative Diseases and Cancer Therapy"

_biomolecules, 2023, doi:10.3390/biom13091388_

Round 1

Reviewer 1 Report

In this article entitled “Cannabinoids in Medicine: A Multifaceted Exploration of Types, Therapeutic Applications, and Emerging Opportunities in Neurodegenerative Diseases and Cancer Therapy” by Voicu et al the authors provide a comprehensive review of the cannabinoids place in modern medicine.

General comments – 

The review is very long and detailed, well over 50 pages in length and over 25.000 words – however it is very instructive, and I do not see a suggested maximum word count in instructions to authors in this journal. It is well over organised with specific information on the background of cannabinoids, and emphasis on the role of endogenous cannabinoids in the body with mechanisms and therapeutic uses also being discussed in how to approach the management of neurodegenerative diseases and cancer in the future. In summary this is a timely and valuable paper containing a comprehensive outline of the key properties of cannabinoids and their promise for the future. 

Specific comments. 

In general, the report is well written with some minor errors.  The Figures and Tables provide a useful graphic summary of certain sections of the text.         

Once a term is abbreviated it should not be necessary to use the full unabbreviated term again – the terms Endocannabinoid System (ECS) and D9-tetrahydrocannabinol (D9-THC) are used a number of times throughout the text and once would be better and use of the abbreviated form thereafter.     

Author Response

Dear reviewers

We would like to, firstly thank you, for the well-thought observations you have made regarding the manuscript and for also taking the time to read our work. We have worked tirelessly to meet all your great points you have made, and we believe that, in the end, the quality of the manuscript has been improved greatly.

The biggest problem- the bibliography, has been addressed thoroughly. We have added to each scientific claim a backing article, increasing the scientific background of our review. Moreover, we have deleted the repetitive statements, rethought paragraphs and corrected misspellings, all of these resulting in a better flow of the text, making it easier to read and understand.

Last but not least, we have addressed the abbreviation problem, and also created a list for the abbreviations which can be found before the bibliography.

All in all, we would like to thank you once again for taking your time to review our work, and for giving such great insights and recommendations that resulted in the improvement of the manuscript’s quality.

With great consideration

The collective of authors. 

Reviewer 2 Report

This is a review article on the medical applications of cannabinoids. This review is very lengthy and certainly contains a lot of information. If it were written in an easy-to-understand manner, it would be a very important review. Unfortunately, however, it can never be said to be easy to read. In addition, many of the references cited are review articles, so the authenticity of the contents cannot be confirmed without further research into the references cited in the cited review. This is very inconvenient. Therefore, the review article in its current form is unacceptable and needs to be newly prepared.

Author Response

(The authors gave the same response as above.)

Round 2

Reviewer 2 Report

Thanks to your immediate and immense efforts, this review has been improved to be very clear and easy to understand. It will surely be a valuable review that will be of great help to readers.

Minor corrections needed

The use of ";" is too frequent. This makes the sentences longer and more difficult to read. The sentences should be separated.

p 7 Fourth line from the end;    [69, p.1] ==> ?

Tables 1, 2, and Figures 1-3 should be shown in the text.

In Table 2, each citation (with corresponding number) should be shown.

I hope these comments are helpful.

Author Response

Dear Reviewer,

Thank you for all of the great suggestions and comments! Following your guidance truly increased the value of the manuscript!

We have addressed the minor concerns.

Best regards and warm wishes!